# pFedMMA: Personalized Federated Fine-Tuning with Multi-Modal Adapter for Vision-Language Models

**Sajjad Ghiasvand, Mahnoosh Alizadeh, & Ramtin Pedarsani**
Department of Electrical and Computer Engineering
UC Santa Barbara
Santa Barbara, CA 93106, USA
{sajjad,alizadeh,ramtin}@ucsb.edu

## Abstract

Vision-Language Models (VLMs) like CLIP have demonstrated remarkable generalization in zero- and few-shot settings, but adapting them efficiently to decentralized, heterogeneous data remains a challenge. While prompt tuning has emerged as a popular parameter-efficient approach in personalized federated learning, existing methods often sacrifice generalization in favor of personalization, struggling particularly on unseen classes or domains. In this work, we propose pFedMMA, a personalized federated learning framework that leverages multi-modal adapters for vision-language tasks. Each adapter contains modality-specific up- and down-projection layers alongside a globally shared projection that aligns cross-modal features. Our optimization strategy allows clients to locally adapt to personalized data distributions while collaboratively training the shared projection to improve global generalization. This design is also communication-efficient, as only the shared component is exchanged during communication rounds. Through extensive experiments across eleven datasets, including domain- and label-shift scenarios, we show that pFedMMA achieves state-of-the-art trade-offs between personalization and generalization, outperforming recent federated prompt tuning methods. Code is available at https://github.com/sajjad-ucsb/pFedMMA.

## 1 Introduction

Vision-Language Models (VLMs) like CLIP Radford et al. (2021) have revolutionized multi-modal learning by jointly embedding visual and textual data through massive contrastive pre-training Jia et al. (2021); Li et al. (2022); Yao et al.. This paradigm empowers models to generalize effectively in zero-shot and few-shot settings Zhang et al. (2022); Zhu et al. (2023); Ghiasvand et al. (2025a); Aghdam & Hu (2025). Among them, larger transformer-based variants Vaswani (2017) (e.g., CLIP ViT-L/14) consistently outperform smaller counterparts such as ViT-B/16, with margins exceeding 6% on benchmarks like ImageNet Deng et al. (2009). However, the computational demands of fine-tuning such large-scale models with billions of parameters pose significant challenges, particularly for domain-specific tasks Oskouie et al. (2025). To mitigate this, Parameter-Efficient Fine-Tuning (PEFT) techniques have emerged, especially in NLP. These methods, including adapters Chen et al. (2022); Karimi Mahabadi et al. (2021); Rebuffi et al. (2017) and prompt tuning Jia et al. (2022); Li & Liang (2021), introduce a lightweight set of trainable parameters or tokens, allowing the backbone model to remain frozen.

While highly effective in centralized settings, these techniques fall short in scenarios involving decentralized and privacy-sensitive data, such as healthcare, legal, or industrial domains Manoel et al. (2023); Shoham & Rappoport (2023); Mahjourian & Nguyen (2025). Federated Learning (FL) offers a promising alternative by enabling collaborative training without raw data sharing. In FL, clients update their local models and transmit only intermediate model updates such as parameters or gradients, which are aggregated into a global model by a central server McMahan et al. (2017).

In real-world scenarios, client data often exhibits variations in domain discrepancies (feature shift) Li et al. or imbalanced class distributions (label shift) Li et al. (2021a). Simply applying standard aggregation strategies, such as FedAvg McMahan et al. (2017), over prompts Guo et al. (2023b) or other fine-tuning methods, such as LoRA, often leads to suboptimal performance due to data heterogeneity Zhang et al. (2023); Borazjani et al. (2025). As a result, Personalized Federated Learning (PFL), particularly with prompt tuning, has gained increasing attention. pFedPrompt Guo et al. (2023a) introduces personalization by coupling a global text prompt with local visual attention modules to tailor predictions to each client's data. FedOTP Li et al. (2024) uses Optimal Transport to align local and global representations under label shift. FedPGP Cui et al. (2024) applies prompt-wise contrastive learning to enhance inter-client generalization. Recently, pFedMoAP Luo et al. (2025) proposes a Mixture-of-Experts framework, where prompts from other clients serve as non-local experts, and each client learns an attention-based gating mechanism for selective adaptation. While these methods achieve impressive personalization performance, they often struggle to generalize to unseen classes or domains, limiting their applicability in out-of-distribution scenarios. For example, as shown in Fig. 1, FedOTP achieves poor harmonic mean accuracy, even though it has been shown to have strong personalization performance.

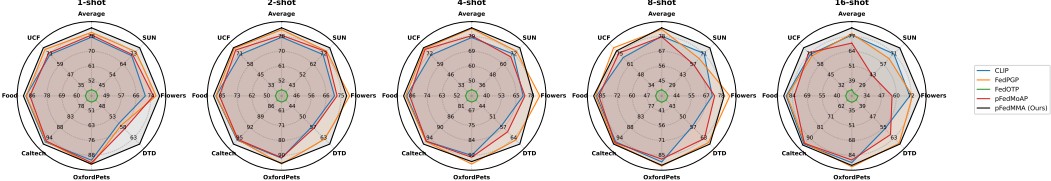

Figure 1: Few-shot performance across datasets using the ViT-B/16 model. Each radar chart illustrates accuracy (%) for a fixed shot count, with spokes representing the evaluation datasets. Curves correspond to different methods, and values increase outward (20–100%). Accuracy is reported as the harmonic mean (HM) over local, base, and novel classes for each dataset and shot.

Beyond prompt tuning, adapters offer another PEFT strategy by introducing small trainable modules into frozen pre-trained models Cai et al. (2020); Chen et al. (2022; b); Gao et al. (2024); Hu et al. (2021); Zhang et al. (2022). Unlike prompts, adapters operate independently of model architecture and can be easily inserted into various backbones, such as ResNets He et al. (2016), ViTs Dosovitskiy et al., and Swin Transformers Liu et al. (2021). However, most adapter methods like AdaptFormer Chen et al. (2022) and LoRA Hu et al. (2021) are uni-modal and do not account for the cross-modal dependencies inherent in VLMs like CLIP Radford et al. (2021). Multi-modal adapters Yang et al. (2024) address this by integrating both visual and textual signals via a shared projection layer that promotes feature alignment across modalities while preserving modality-specific knowledge. Despite their demonstrated advantages over prompt-based approaches Yang et al. (2024); Guo & Gu (2025), their integration with PFL remains largely unexplored.

In this work, we introduce a **P**ersonalized **F**ederated **M**ulti-**M**odal **A**dapter (**pFedMMA**) architecture that adopt a multi-modal adapter design with three components: a modality-specific down-projection, a shared projection, and a modality-specific up-projection. During training, all components are updated locally by each client, but only the shared projection is globally aggregated. *This asymmetric training scheme enables effective personalization through client-specific projections, while promoting generalization via a shared modality-alignment space.* Moreover, since only the shared adapter is communicated during rounds, the method remains communication-efficient. As confirmed by our experiments, this design achieves the strongest trade-off between personalization and generalization under both feature and label shifts. As shown in Fig. 1, on average, our proposed pFedMMA delivers the best harmonic mean performance compared to state-of-the-art federated prompt tuning methods.

Before delving into details, we summarize our contributions: **(1)** We observe that while most state-of-the-art prompt tuning methods achieve strong personalization performance, they often generalize poorly to unseen classes. To address this, we introduce a multi-modal adapter framework that explicitly aims to balance personalization and generalization in federated vision-language learning. **(2)** We propose pFedMMA, an adapter-based approach for PFL of VLMs. Our architecture incorporates modality-specific up- and down-projection layers and a shared cross-modal projection. All components are updated locally, but only the shared projection is aggregated globally, enabling

effective asymmetric optimization. **(3)** We conduct extensive experiments on widely used benchmarks to evaluate pFedMMA's performance on base-to-novel generalization across both category- and domain-level tasks under heterogeneous data distributions. Results demonstrate the superiority of our approach in harmonizing generalization and personalization.

## 2 PRELIMINARIES

### 2.1 PERSONALIZED FEDERATED LEARNING

Traditional federated learning frameworks are designed around the principle of global consensus, where the goal is to collaboratively train a single model that generalizes well across a federation of clients. The canonical approach, FedAvg McMahan et al. (2017), formalizes this as the minimization of a weighted average of local objectives: $\min_{\boldsymbol{\theta}} F(\boldsymbol{\theta}) = \sum_{i=1}^{N} p_i F_i(\boldsymbol{\theta})$, where $\boldsymbol{\theta}$ denotes the global model, $F_i(\cdot)$ represents the local empirical loss of client $i$, and $p_i = \frac{n_i}{n}$ scales the contribution of each client by its dataset size $n_i$, with $n = \sum_i n_i$. In this setup, each client's local loss is computed as the average over its data: $\sum_{k=1}^{n_i} \mathcal{L}_i(\boldsymbol{\theta} \mid (\boldsymbol{x}_k, y_k))$, where $\mathcal{L}_i$ is the local loss function and $(\boldsymbol{x}_k, y_k)$ is the $k$-th data point on client $i$.

In contrast, personalized federated learning (PFL) challenges the one-size-fits-all paradigm by allowing each client to maintain its own model $\boldsymbol{\theta}_i$. This formulation acknowledges data heterogeneity and aims to tailor learning to each client's unique distribution. The objective for PFL becomes:

$$\min_{\boldsymbol{\theta}_1, \ldots, \boldsymbol{\theta}_N} F(\boldsymbol{\theta}_1, \ldots, \boldsymbol{\theta}_N) = \sum_{i=1}^{N} p_i F_i(\boldsymbol{\theta}_i), \tag{1}$$

offering a flexible alternative that prioritizes personalized performance over strict global consensus.

### 2.2 VISION-LANGUAGE CLASSIFICATION WITH FEW-SHOT ADAPTATION

In vision-language classification, predictions emerge from the powerful alignment between visual and textual modalities established during pretraining. Given a label set with $K$ classes, the model begins by crafting natural language prompts Liu et al. (2023)—semantic descriptions like "a photo of a [class name]"—for each class $c_k$. These textual cues are passed through a frozen text encoder $\theta_t$, producing normalized text embeddings $\mathbf{z}_k^{(T)} = \theta_t(c_k) \in \mathbb{R}^d$. In parallel, each input image $\boldsymbol{x}_i$ is processed by a visual encoder $\theta_v$, generating a corresponding normalized image embedding $\mathbf{z}_i^{(I)} = \theta_v(\boldsymbol{x}_i) \in \mathbb{R}^d$. Classification then hinges on comparing the cosine similarity between these multimodal representations. The result is a set of logits transformed into class probabilities via a temperature-scaled softmax: $p_{i,k} = \exp\left(\cos(\mathbf{z}_i^{(I)}, \mathbf{z}_k^{(T)})/\gamma\right) / \sum_{j=1}^{K} \exp\left(\cos(\mathbf{z}_i^{(I)}, \mathbf{z}_j^{(T)})/\gamma\right)$, where $\gamma$ is the temperature parameter controlling distribution sharpness. The predicted label for image $\mathbf{x}_i$ corresponds to the class with the highest posterior probability: $\hat{k} = \arg\max_k p_{i,k}$.

This zero-shot classification pipeline mirrors the contrastive training strategy employed in foundational vision-language models like CLIP Radford et al. (2021), enabling impressive generalization to novel tasks without requiring any target-domain fine-tuning.

To further tailor the model to downstream tasks, the few-shot setting introduces a small set of labeled examples per class, typically fewer than 16. With $M$ support samples per class and ground-truth labels encoded as one-hot vectors $y_{ik}$ (where $y_{ik} = 1$ if $\mathbf{x}_i$ belongs to class $k$, and 0 otherwise), classification proceeds identically to the zero-shot case. However, the model is now adapted by minimizing the cross-entropy loss over the labeled support set: $\mathcal{L}_{\text{CE}} = -\frac{1}{M} \sum_{i=1}^{M} \sum_{k=1}^{K} y_{ik} \ln p_{i,k}$.

This fine-tuning step enables the model to better capture domain-specific semantics while maintaining the efficiency and generalization capabilities of the pretrained architecture. Adaptation can be achieved through various strategies. One approach is to directly optimize the input prompts $\{c_k\}_{k=1}^{K}$, following the principles of prompt tuning Chen et al. (a). Alternatively, lightweight task-specific modules such as adapter layers Gao et al. (2024) or low-rank parameterizations like LoRA Zanella & Ben Ayed (2024) can be fine-tuned, while keeping the backbone encoders frozen.

## 2.3 Fine-Tuning via Parallel Adapters

In contrast to the serial adapter architecture introduced by Houlsby et al. (2019), where adapter modules are inserted sequentially after each sub-layer (e.g., attention or feed-forward), *parallel adapters* He et al. adopt an alternative integration strategy. Rather than placing the adapter transformation after the main layer, the parallel formulation processes the input through the adapter module concurrently with the frozen backbone transformation and combines their outputs additively.

Let $x \in \mathbb{R}^d$ be the input to a transformer sub-layer, and let $f(x)$ denote the frozen pre-trained transformation. A parallel adapter layer computes the output as: $\text{Output}(x) = f(x) + \alpha A(x)$, where $\alpha$ is a scaling factor and the adapter module $A(x)$ uses the same bottleneck structure as in the serial configuration: $A(x) = U(\delta(D(x)))$, where $U$ is an up-projection affine map, $D$ is a down-projection affine map, and $\delta$ is a non-linear activation function such as ReLU. If the input $x$ has dimensionality $d$, then $D \in \mathbb{R}^{r \times d}$ and $U \in \mathbb{R}^{d \times r}$, where $r \ll d$. This bottleneck structure introduces significantly fewer trainable parameters compared to the full model. As with serial adapters, only the adapter parameters are trained during fine-tuning, and the base model remains frozen. Parallel adapters preserve model expressiveness while enabling efficient adaptation with minimal architectural modifications.

## 3 Proposed Method

In this section, we introduce pFedMMA, a novel framework that leverages multi-modal adapters to efficiently and effectively adapt large pre-trained VLMs under federated learning settings. Our design consists of two central components: (i) a multi-modal adapter architecture that bridges and enriches representations across visual and textual modalities, and (ii) a hybrid personalization strategy that promotes both generalization and personalization by decoupling local and shared adapter components.

### 3.1 Multi-Modal Adapter Architecture

We build on the adapter-based design introduced in Yang et al. (2024) to incorporate a lightweight and efficient tuning mechanism for vision-language models. This architecture has proven effective in few-shot generalization settings, where pre-trained CLIP models are fine-tuned on a limited number of base classes and tested on base and novel, unseen categories.

The motivation for this design stems from two empirical findings: (i) higher layers of both image and text encoders in CLIP contain more discriminative and dataset-specific features, while lower layers preserve general, transferable knowledge; and (ii) larger modality gaps between text and image encoders are observed in the lower layers, making cross-modal alignment particularly challenging in the early stages of the network Yang et al. (2024).

Based on these insights, the multi-modal adapter is inserted into the upper transformer blocks of both encoders, starting from block $\ell$, while the lower layers remain frozen. This helps preserve general representations while enabling task-specific adaptation at the top layers.

Each adapter consists of: **(i)** A **down-projection** layer that reduces the input dimension, **(ii)** A **shared projection** layer that facilitates interaction between the modalities, **(iii)** An **up-projection** layer that restores the original dimension.

This three-part structure allows the adapter to first transform features into a low-dimensional space, fuse them through a shared module, and then project them back. Formally, for the visual adapter (indexed by $(I)$) and the textual adapter (indexed by $(T)$) at the $j$-th block:

$$\mathcal{A}_j^{(o)}(z_j^{(o)}) = W_{ju}^{(o)} \cdot \delta(W_{js} \cdot \delta(W_{jd}^{(o)} \cdot z_j^{(o)})), \quad o \in \{I, T\}, \quad j \in \{\ell, \cdots, L\}, \qquad (2)$$

where $z_j^{(I)}$ and $z_j^{(T)}$ denote the input hidden states at the $j$-th transformer layer for the vision and text encoders, $W_{jd}^{(I)}$ and $W_{jd}^{(T)}$ are the *down-projection* matrices, $W_{js}$ is the *shared projection* matrix used across both modalities, $W_{ju}^{(I)}$ and $W_{ju}^{(T)}$ are the *up-projection* matrices, and $\delta(\cdot)$ denotes the non-linear activation function (e.g., GELU), applied element-wise.

This shared projection structure encourages information exchange across modalities, while still maintaining modality-specific processing through separate up/down projections.

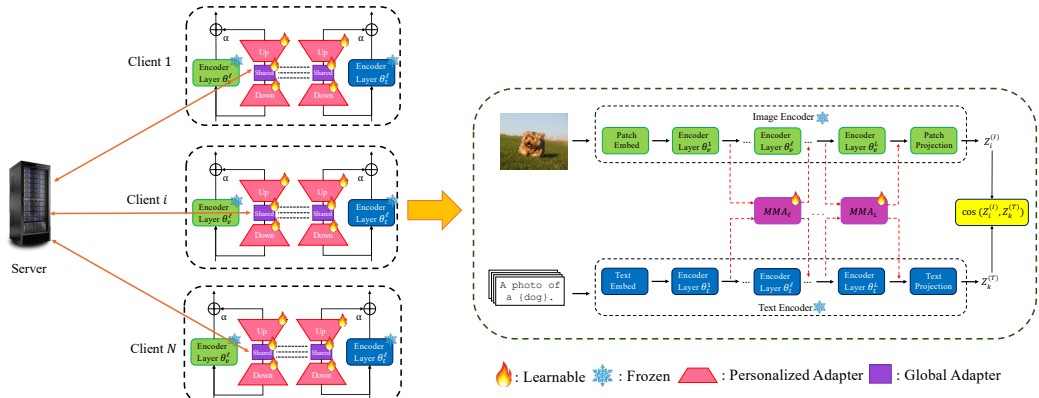

Figure 2: An overview of the pFedMMA framework. Each client independently updates all trainable components of the multi-modal adapters including client-specific up/down projections and the shared projection over local epochs. After local training, only the shared adapter is uploaded and aggregated by the server. This design promotes personalization through local adapters while enabling generalization via a globally shared component.

In contrast to methods that inject prompts or adapters across all layers Chen et al. (2022); Houlsby et al. (2019); Hu et al. (2021); Ghiasvand et al. (2025b) or some lower layers Khattak et al. (2023a;b); Zhou et al. (2022b;a), this selective, top-layer insertion strategy reduces the number of trainable parameters while maintaining cross-modal adaptability.

## 3.2 Generalization and Personalization via pFedMMA

To effectively balance generalization and personalization in federated vision-language learning, we propose a hybrid training strategy within the multi-modal adapter framework. Each adapter consists of three projection components: a modality-specific *down-projection*, a *shared projection*, and a modality-specific *up-projection*. In our personalization scheme, clients update the down- and up-projection components locally, while the shared projection matrix is synchronized globally via server aggregation.

This selective update mechanism provides several key benefits: **(i) Local personalization:** By allowing clients to optimize their own up- and down-projection matrices, each client can adapt the representation space to their unique local data distribution. This is particularly effective under label and feature heterogeneity. **(ii) Global generalization:** The shared projection matrix is collaboratively trained across clients and is responsible for aligning the modalities in a consistent global space. This facilitates transferability and enables the model to generalize well across diverse domains and tasks. **(iii) Communication efficiency:** Since the shared projection layer is low-dimensional compared to the full model or full adapter stack, transmitting only the shared component during communication rounds results in significantly reduced communication cost. Extensive communication and computational cost analysis are provided in Appendix E.

Specifically, for a client $i$ in communication round $t$, all trainable parameters

$$\boldsymbol{W} \in \left\{ \boldsymbol{W}_{jd,i}^{(I)}, \boldsymbol{W}_{ju,i}^{(I)}, \boldsymbol{W}_{jd,i}^{(T)}, \boldsymbol{W}_{ju,i}^{(T)}, \boldsymbol{W}_{js,i} \right\}, \quad j \in \{\ell, \cdots, L\}, \quad i \in \{1, \cdots, N\}, \quad (3)$$

are updated for $E$ local epochs using gradient descent: $\boldsymbol{W}_i^{t,e} = \boldsymbol{W}_i^{t,e-1} - \eta \nabla \mathcal{L}_{ce}(\boldsymbol{W}_i^{t,e-1})$, where $\eta$ is the learning rate, $\mathcal{L}_{ce}$ is the cross-entropy loss, and $e \in \{1, \cdots, E\}$.

After local updates, only the shared projection parameters $\boldsymbol{W}_{js,i}^{t,E}$ are uploaded to the server. These are aggregated across all participating clients to obtain the updated global shared adapter: $\boldsymbol{W}_{js}^{t+1} = \sum_{i=1}^{N} p_i \boldsymbol{W}_{js,i}^{t,E}$, where $p_i = \frac{n_i}{n}$ scales the contribution of each client by its dataset size $n_i$, with $n = \sum_i n_i$ and $N$ is the number of participating clients. In contrast, the up- and down-projection parameters remain local and are not shared or averaged.

This asymmetric update design enables pFedMMA to effectively capture both shared and client-specific information, resulting in an improved balance between personalization and generalization, as demonstrated in our experiments on tasks involving domain and label shifts. The overall training and communication flow of pFedMMA is illustrated in Fig 2.

## 4 EMPIRICAL RESULTS

In this section, we conduct extensive experiments to evaluate the generalization and personalization capability of pFedMMA in heterogeneous data distribution scenarios.

### 4.1 EXPERIMENTAL SETUP

**Datasets and Data Heterogeneity.** To evaluate the effectiveness of pFedMMA, we conduct experiments across eleven public benchmark datasets that cover various types of data heterogeneity, including label shift and feature shift. Following prior work such as Guo et al. (2023b), we use seven visual classification datasets: SUN397 Xiao et al. (2010), OxfordPets Parkhi et al. (2012), Flowers102 Nilsback & Zisserman (2008), DTD Cimpoi et al. (2014), Caltech101 Fei-Fei et al. (2004), UCF101 Soomro et al. (2012), and Food101 Bossard et al. (2014). We refer to these as the CLIP datasets. To simulate severe label heterogeneity, we apply a pathological non-IID setting in which each client is assigned a distinct, non-overlapping set of classes. Clients are trained on their *local* classes and evaluated on their own classes, on *base* classes held by other clients, and on *novel* classes that are unseen during the training process.

To evaluate performance under feature shift, we utilize two widely adopted multi-domain datasets: DomainNet Peng et al. (2019), which consists of six distinct domains, and Office-Caltech10 Gong et al. (2012), which includes four domains. Following prior studies, each client is assigned data from a single domain, ensuring that every domain is represented by a group of clients in the federation. To introduce additional heterogeneity and simulate realistic federated learning scenarios, we further partition the data within each domain using a symmetric Dirichlet distribution with concentration parameter $\beta$. This setup introduces both feature shift across domains and label shift within domains. All domains participate in both training and evaluation phases, allowing us to assess cross-domain generalization and personalization performance in more realistic federated conditions.

For personalization evaluation, we include CIFAR-10 Krizhevsky et al. (2010) and CIFAR-100 Krizhevsky et al. (2009). These datasets are partitioned among clients using a Dirichlet distribution, which creates varying degrees of label skew across clients. Additionally, we apply the same pathological class split as used in the CLIP datasets to test robustness under extreme heterogeneity. Further details on the dataset configurations and partitioning strategies can be found in Appendix C.1.

**Baselines.** We evaluate pFedMMA across all experimental settings, including generalization, personalization, and domain generalization, using a consistent set of five baselines. Zero-shot CLIP Radford et al. (2021) serves as a non-adaptive reference model that uses fixed hand-crafted prompt templates such as "a photo of a [class]" without any task-specific learning. PromptFL Guo et al. (2023b) represents a standard federated prompt learning approach in which a shared prompt is collaboratively learned across clients using FedAvg. FedPGP Cui et al. (2024) introduces prompt-wise contrastive learning to encourage consistency between global and local prompts. FedOTP Li et al. (2024) applies unbalanced Optimal Transport to align global knowledge with client-specific prompt representations. Finally, pFedMoAP Luo et al. (2025) leverages a Mixture-of-Experts design that enables each client to access both local and non-local prompt experts through an attention-based gating mechanism. In addition to these prompt-based methods, we also consider adapter and LoRA-style PEFT baselines by implementing federated CLIP-Adapter Gao et al. (2024) and federated CLIP-LoRA Zanella & Ben Ayed (2024), where only the adapter or low-rank layers are updated and aggregated across clients. These baselines cover a diverse range of federated adaptation strategies, providing a strong benchmark for assessing the performance of pFedMMA across different types of heterogeneity.

**Implimentation Details.** All methods, including pFedMMA and all baselines, are implemented on top of a frozen CLIP model. We use two backbone architectures, ViT-B16 and ViT-B32 Dosovitskiy et al., and default to ViT-B16 unless otherwise specified. For the CLIP datasets, each is split into 10 clients with non-overlapping classes, using 100 percent participation, 2 local epochs, and 50 communication rounds. For the CIFAR-10 and CIFAR-100 datasets, we simulate a large-scale

Table 1: Top-1 accuracy (%) of different methods across 7 datasets in the 16-shot setting.

| Method | Average on 7 datasets | | | | SUN397 | | | | Flowers102 | | | | DTD | | | |
|---|---|---|---|---|---|---|---|---|---|---|---|---|---|---|---|---|
| | Local | Base | Novel | HM | Local | Base | Novel | HM | Local | Base | Novel | HM | Local | Base | Novel | HM |
| CLIP Radford et al. (2021) | 76.36 | 76.81 | 81.21 | 78.03 | 69.41 | 69.38 | 75.52 | 71.32 | 67.89 | 69.23 | 76.88 | 71.12 | 54.26 | 54.86 | 59.18 | 56.02 |
| PromptFL Guo et al. (2023b) | 88.93 | 88.95 | 75.36 | 83.09 | 77.73 | 77.71 | 72.96 | 76.07 | 97.37 | 97.06 | 63.62 | 82.66 | 80.23 | 80.21 | 45.29 | 63.81 |
| FedCLIP-Adapter Gao et al. (2024) | 78.97 | 79.22 | 82.09 | 80.04 | 72.82 | 72.78 | 77.22 | 74.22 | 71.46 | 73.22 | 77.38 | 73.94 | 56.99 | 55.90 | 59.78 | 57.51 |
| FedCLIP-LoRA Zanella & Ben Ayed (2024) | 81.69 | 90.16 | 79.76 | 83.35 | 74.67 | 80.46 | 72.28 | 75.65 | 76.77 | 97.72 | 68.16 | 79.09 | 68.16 | 81.94 | 59.42 | 68.64 |
| FedPGP Cui et al. (2024) | 95.38 | 76.49 | 71.68 | 79.09 | 94.29 | 54.88 | 57.76 | 65.02 | 99.67 | 72.44 | 58.65 | 73.37 | 89.03 | 71.03 | 50.94 | 66.75 |
| FedOTP Li et al. (2024) | 97.34 | 18.00 | 36.69 | 31.08 | 94.50 | 11.51 | 14.86 | 18.21 | 99.65 | 14.62 | 30.49 | 26.97 | 98.08 | 20.79 | 35.36 | 34.65 |
| pFedMoAP Luo et al. (2025) | 97.89 | 61.82 | 66.60 | 71.05 | 95.93 | 31.18 | 35.40 | 42.41 | 99.81 | 43.70 | 48.37 | 55.99 | 96.43 | 53.60 | 48.21 | 60.28 |
| pFedMMA (Ours) | 97.17 | 77.40 | 81.49 | 84.15 | 94.06 | 70.99 | 76.37 | 79.34 | 95.58 | 71.54 | 76.00 | 79.79 | 97.45 | 55.44 | 61.55 | 67.35 |
| Δ | −0.74% | +1.19% | +13.69% | +6.4% | −1.95% | +29.35% | +32.22% | +22.02% | −4.24% | −1.24% | +29.58% | +8.75% | −0.64% | −21.95% | +20.83% | +0.9% |

| Method | OxfordPets | | | | Caltech101 | | | | Food101 | | | | UCF101 | | | |
|---|---|---|---|---|---|---|---|---|---|---|---|---|---|---|---|---|
| | Local | Base | Novel | HM | Local | Base | Novel | HM | Local | Base | Novel | HM | Local | Base | Novel | HM |
| CLIP Radford et al. (2021) | 89.45 | 89.42 | 96.81 | 91.77 | 96.14 | 97.22 | 94.21 | 95.84 | 89.40 | 89.42 | 90.70 | 89.84 | 68.00 | 68.15 | 75.18 | 70.29 |
| PromptFL Guo et al. (2023b) | 96.35 | 96.28 | 97.26 | 96.63 | 97.77 | 98.19 | 92.58 | 96.11 | 90.48 | 90.50 | 91.37 | 90.78 | 82.57 | 82.73 | 64.47 | 75.55 |
| FedCLIP-Adapter Gao et al. (2024) | 93.01 | 92.93 | 97.09 | 94.30 | 96.28 | 97.35 | 94.10 | 95.89 | 90.07 | 90.10 | 91.19 | 90.45 | 72.13 | 72.23 | 77.88 | 73.98 |
| FedCLIP-LoRA Zanella & Ben Ayed (2024) | 90.71 | 95.64 | 97.71 | 94.59 | 96.76 | 97.93 | 94.43 | 96.35 | 89.18 | 90.32 | 91.49 | 90.32 | 75.58 | 87.13 | 74.80 | 78.79 |
| FedPGP Cui et al. (2024) | 96.62 | 95.17 | 97.15 | 96.31 | 99.42 | 94.94 | 90.88 | 94.95 | 93.70 | 86.38 | 87.14 | 88.96 | 94.94 | 60.62 | 59.26 | 68.33 |
| FedOTP Li et al. (2024) | 100.00 | 11.60 | 51.22 | 25.92 | 99.94 | 36.47 | 62.77 | 56.23 | 95.69 | 17.29 | 37.97 | 31.70 | 93.54 | 13.62 | 24.19 | 23.91 |
| pFedMoAP Luo et al. (2025) | 99.92 | 77.61 | 92.05 | 88.87 | 99.92 | 94.07 | 92.43 | 95.37 | 97.49 | 69.86 | 83.51 | 82.09 | 95.79 | 62.74 | 66.23 | 72.33 |
| pFedMMA (Ours) | 100.00 | 88.50 | 96.60 | 94.78 | 100.00 | 96.53 | 94.29 | 96.88 | 97.45 | 89.15 | 90.77 | 92.32 | 95.63 | 69.61 | 74.88 | 78.58 |
| Δ | 0% | −7.01% | −0.57% | −1.59% | +0.06% | +1.67% | +2.01% | +1.58% | −0.04% | +3.21% | +4.17% | +3.78% | −0.17% | +10.95% | +13.06% | +8.64% |

Figure 3: Local and harmonic mean (HM) accuracies of various methods across different shot settings.

federated environment with 100 clients, using a varying Dirichlet distribution and a 10 percent client participation rate per communication round. Training runs for 50 rounds with 1 local epoch per round. In the case of DomainNet and Office-Caltech10, each domain of these two datasets is partitioned to 1/2 clients, resulting in $N = 6/12$ for DomainNet and $N = 4/8$ for Office-Caltech10. We use SGD with a learning rate of 0.001, and batch sizes of 32 for training and 100 for testing. Further implementation details are provided in the Appendix, where we also report additional results using the AdamW optimizer (Table 19), which exhibit similar trends to those obtained with SGD.

## 4.2 PERFORMANCE EVALUATION

**Base-to-Novel Class Generalization.**

We evaluate the performance of pFedMMA in terms of its ability to generalize from locally trained classes to both base and novel classes. Following prior work, we report top-1 accuracy on each client's local classes, on the base classes seen by other clients, and on novel classes that are entirely unseen during training. To capture overall effectiveness, we use the harmonic mean (HM) of these three metrics, $HM = 3/(Acc_{local}^{-1} + Acc_{base}^{-1} + Acc_{novel}^{-1})$, which penalizes methods that over-optimize one component at the expense of the others and thus better reflects the balance between personalization (local) and generalization (base and novel) than a simple arithmetic mean; this type of harmonic-mean score is standard in generalized zero-shot learning and base-to-novel CLIP adaptation, and has also been adopted in recent PFL work to jointly summarize local, base, and novel accuracies Verma et al. (2020); Du et al. (2025); Cui et al. (2024). As summarized in Table 1 for the 16-shot setting, pFedMMA consistently achieves strong performance across all evaluation categories and delivers the best overall HM averaged across seven datasets, outperforming all baselines.

Zero-shot CLIP, PromptFL, federated CLIP-Adapter, and CLIP-LoRA suffer from poor local accuracy, tending to favor generalization at the expense of personalization. We also report Δ, which denotes the relative improvement of pFedMMA compared with the strongest non-baseline methods (FedPGP, FedOTP, and pFedMoAP). While FedOTP sometimes achieves high local accuracy, its extremely low

Table 2: Accuracy comparison (%) on the Dirichlet Non-IID setting in CIFAR-10 and CIFAR-100.

| Dataset | CIFAR-100 | | | | | | CIFAR-10 | | | | | |
|---|---|---|---|---|---|---|---|---|---|---|---|---|
| #$\beta$ | 0.1 | 0.3 | 0.5 | 1 | 5 | 10 | 0.1 | 0.3 | 0.5 | 1 | 5 | 10 |
| CLIP Radford et al. (2021) | 64.93 | 64.90 | 65.00 | 64.95 | 64.94 | 64.91 | 87.98 | 87.95 | 87.93 | 87.98 | 88.02 | 87.98 |
| PromptFL Guo et al. (2023b) | 75.34 | 73.48 | 72.85 | 72.83 | 72.81 | 72.41 | 92.80 | 92.95 | 94.34 | 93.89 | 93.31 | 93.02 |
| FedPGP Cui et al. (2024) | 74.72 | 72.89 | 74.85 | 74.18 | 74.07 | 73.90 | 91.69 | 93.19 | 93.21 | 92.98 | 93.04 | 92.91 |
| FedOTP Li et al. (2024) | 77.53 | 73.83 | 72.21 | 70.99 | 69.40 | 68.97 | 97.23 | 95.82 | 94.64 | 93.10 | 91.87 | 91.67 |
| pFedMoAP Luo et al. (2025) | 80.29 | 75.70 | 75.68 | 74.53 | 73.00 | 72.61 | 97.13 | 95.92 | 94.86 | 93.97 | 92.67 | 92.65 |
| pFedMMA (Ours) | 81.82 | 78.33 | 76.92 | 75.70 | 74.03 | 73.65 | 97.37 | 96.92 | 95.82 | 94.82 | 93.52 | 93.07 |
| $\Delta$ | +1.91% | +3.47% | +1.64% | +1.57% | −0.05% | −0.34% | +0.14% | +1.04% | +1.01% | +0.9% | +0.23% | +0.05% |

Table 3: Test accuracy (%) of different methods on DomainNet and Office-Caltech10 with lable shift and domain shift using Dirichlet partitioning ($\beta = 0.5$).

| Method | DomainNet | | | | | | | Office-Caltech10 | | | | |
|---|---|---|---|---|---|---|---|---|---|---|---|---|
| | Clipart | Infograph | Painting | Quickdraw | Real | Sketch | Avg. | Amazon | Caltech | DSLR | Webcam | Avg. |
| CLIP Radford et al. (2021) | 8.99 | 10.69 | 11.20 | 10.85 | 9.53 | 9.39 | 10.11 | 11.78 | 6.21 | 9.92 | 8.51 | 9.11 |
| PromptFL Guo et al. (2023b) | 11.02 | 1.65 | 11.20 | 8.95 | 13.89 | 20.75 | 11.24 | 10.35 | 15.83 | 32.06 | 7.13 | 16.34 |
| FedPGP Cui et al. (2024) | 24.77 | 31.87 | 23.87 | 22.87 | 22.40 | 23.64 | 24.90 | 20.34 | 19.12 | 20.85 | 22.52 | 20.71 |
| pFedMoAP Luo et al. (2025) | 24.77 | 30.93 | 26.09 | 20.46 | 22.59 | 23.10 | 24.65 | 20.01 | 24.45 | 18.02 | 15.73 | 19.55 |
| pFedMMA (Ours) | 50.38 | 23.81 | 60.27 | 61.44 | 40.35 | 46.79 | 47.17 | 9.26 | 29.15 | 33.26 | 13.64 | 21.33 |

base and novel class scores indicate poor generalization. pFedMoAP performs well on local classes due to its MoE-based prompt sharing, but it lags behind pFedMMA in base and novel accuracy. By contrast, pFedMMA achieves the highest base and novel accuracy, surpassing FedPGP and demonstrating excellent generalization, while remaining competitive on local classes—only $0.74\%$ lower than pFedMoAP.

Fig. 3 illustrates local and HM accuracy across varying numbers of shots $\{1, 2, 4, 8, 16\}$, showing the same performance pattern. Detailed results for all datasets are provided in Table 9 in the Appendix.

**Evaluation on Personalization.** We further evaluate the personalization capability of pFedMMA on CIFAR-10 and CIFAR-100 under a challenging Dirichlet partitioning scheme, varying the concentration parameter $\beta$ across 100 clients with 10% client participation per communication round. The results, summarized in Table 2, show that pFedMMA consistently achieves the highest accuracy on both datasets, demonstrating its strong adaptability to highly non-IID data distributions.

**Model Evaluation on Feature & Label Shifts.** To evaluate the robustness of pFedMMA in realistic federated learning scenarios, we examine its performance under both label shift and feature shift using the DomainNet and Office-Caltech10 datasets. Following the standard protocol, each domain is split into two clients via a Dirichlet distribution with $\beta = 0.5$, yielding 12 clients for DomainNet and 8 clients for Office-Caltech10. The results in Table 3 show that under these challenging heterogeneous conditions, traditional methods such as CLIP and PromptFL struggle to generalize effectively. In contrast, pFedMMA consistently achieves the highest average accuracy across both datasets, highlighting its strong robustness to cross-domain shifts. Additional experiments with one or two clients per domain and varying $\beta$ are provided in Tables 11 and 12 in the Appendix.

## 4.3 ABLATION STUDY & HYPERPARAMETER SENSITIVITY

**Impact of model.** To further examine the performance of pFedMMA under a different backbone, we report results with ViT-B/32 on the average of six datasets across five shot settings, comparing against three advanced baselines (Table 4). While pFedMMA shows slightly lower local accuracy than FedOTP and pFedMoAP, this gap narrows as the number of shots increases. Importantly, pFedMMA consistently achieves the best trade-off between personalization and generalization, demonstrating stable improvements in the harmonic mean across all settings. Detailed results for all datasets are provided in Table 10 in the Appendix.

**Dimension of the Shared Layer.** Table 5 (bottom-left) reports the average accuracies over four datasets and five shot settings. As shown, using a larger 128-dimensional representation yields slightly better performance than 32 dimensions. However, to keep the number of trainable parameters low, we consistently adopt the 32-dimensional setting throughout the paper. Detailed results are provided in Tables 13 and 14 in the Appendix.

**Scaling Factor $\alpha$.** The scaling factor controls the balance between general features and task-specific features. We systematically evaluate its effect, with results summarized in Table 5 (top-left). Our

Table 4: Average performance across six datasets using the ViT-B/32 backbone under different shot settings (1, 2, 4, 8, and 16).

| Method | 1 Shot | | | | 2 Shots | | | | 4 Shots | | | | 8 Shots | | | | 16 Shots | | | |
|---|---|---|---|---|---|---|---|---|---|---|---|---|---|---|---|---|---|---|---|---|
| | Local | Base | Novel | HM | Local | Base | Novel | HM | Local | Base | Novel | HM | Local | Base | Novel | HM | Local | Base | Novel | HM |
| FedPGP | 79.31 | **79.35** | 80.25 | **79.53** | 82.29 | **81.43** | 77.08 | 80.07 | 85.86 | **81.91** | 74.50 | 79.68 | 89.07 | **80.33** | 73.61 | 79.63 | 93.57 | 70.79 | 68.89 | 74.66 |
| FedOTP | 86.84 | 11.20 | 20.66 | 19.50 | 88.69 | 11.29 | 23.12 | 20.04 | 91.51 | 11.02 | 21.19 | 19.33 | 92.44 | 9.95 | 19.39 | 17.25 | 94.60 | 9.97 | 16.74 | 16.46 |
| pFedMoAP | **93.18** | 45.53 | 54.08 | 56.99 | **95.05** | 47.14 | 58.33 | 59.37 | **96.53** | 44.14 | 54.34 | 56.26 | **96.88** | 42.53 | 50.93 | 53.83 | **97.21** | 29.83 | 44.15 | 44.86 |
| pFedMMA (Ours) | 82.14 | 76.78 | 79.35 | 79.31 | 84.40 | 77.04 | 79.34 | 80.09 | 86.82 | 76.85 | 79.32 | 80.67 | 88.07 | 76.79 | 79.01 | 80.83 | 90.02 | 76.57 | 79.75 | 81.29 |
| Δ | | | | −0.28% | | | | +0.02% | | | | +1.24% | | | | +1.51% | | | | +8.88% |

Table 5: Ablation study on pFedMMA design choices, including scaling factor, adapter dimension, starting layer, and adapter sharing strategies.

| $\alpha$ | Local | Base | Novel | HM |
|---|---|---|---|---|
| 0.0001 | 91.40 | 5.06 | 5.76 | 7.62 |
| 0.0005 | 91.45 | 46.34 | 59.16 | 59.15 |
| 0.001 | 90.91 | 72.21 | 78.48 | 79.37 |
| 0.005 | 91.03 | **78.65** | **81.77** | **83.25** |
| 0.01 | **91.47** | 78.12 | 81.56 | 82.03 |

| $\ell \to L$ | Local | Base | Novel | HM |
|---|---|---|---|---|
| 12 | 96.49 | 76.85 | 81.37 | 83.57 |
| 10 → 12 | **96.61** | 78.14 | **81.98** | **84.38** |
| 8 → 12 | 95.75 | 78.43 | 81.82 | 84.27 |
| 6 → 12 | 91.53 | 78.53 | 81.76 | 83.32 |
| 5 → 12 | 91.58 | **78.67** | 81.71 | 83.38 |

| Dims | Local | Base | Novel | HM |
|---|---|---|---|---|
| 8 | 89.15 | 72.23 | 76.41 | 78.24 |
| 16 | 89.84 | 72.51 | 77.36 | 78.93 |
| 32 | 90.91 | 72.21 | **78.33** | 79.37 |
| 64 | 91.55 | 71.37 | 78.31 | 79.15 |
| 128 | **91.78** | **72.55** | 78.23 | **79.68** |

| Method | DTD | Caltech | Flowers | OxfordPets |
|---|---|---|---|---|
| Baseline 1 | 61.10 | 96.61 | 73.82 | 91.91 |
| Baseline 2 | 62.19 | 98.14 | 77.04 | 92.75 |
| pFedMMA | **76.38** | **99.48** | **86.34** | **97.39** |

pFedMMA achieves the best trade-off performance (HM) between local, base, and novel classes at $\alpha = 0.005$. A larger scaling factor enables faster adaptation to base classes but leads to weaker performance on novel and base classes, whereas a smaller scaling factor hinders effective tuning for downstream tasks. Detailed results are provided in Table 16 in the Appendix.

**Starting Layer $\ell$.** We evaluate different choices of encoder layers for integrating pFedMMA in Table 5 (top-right). As shown, updating the last three layers yields the best HM performance, which we attribute to the limited amount of training data in few-shot settings. Accordingly, we consistently set $\ell = 10$ for CLIP datasets throughout the paper. For other datasets, updating additional layers leads to better results, so we adopt $\ell = 5$. Detailed results are provided in Table 17 in the Appendix.

**Adapting Variant Options for Personalization.** We evaluate the effectiveness of different design choices of MMA in personalized federated learning. In Table 5 (bottom-right), we compare two alternative baselines: treating all adapters as global (Baseline 1) and using the shared adapter as the personalized component while treating the up- and down-projection adapters as global (Baseline 2). As shown, pFedMMA achieves significantly higher local accuracy than both baselines. Moreover, it achieves superior base and novel performance compared to state-of-the-art prompt learning methods, as shown earlier, underscoring its ability to strike a strong balance between personalization and generalization.

**Adapting Variant Options for FL Aggregation.** We next ablate how the shared adapter is aggregated across clients to localize the main information-sharing channel. In Table 6, we compare three variants that differ in which modality-specific shared block is federated: *Vision Only*, where only the vision-side shared block is aggregated; *Text Only*, where only the text-side shared block is aggregated; and *Both Vision & Text*, where separate shared blocks for each modality are aggregated simultaneously. These variants achieve very similar local accuracy, with

Table 6: Comparison of FL aggregation variants (vision-only, text-only, and both-sides) for the shared adapter.

| Methods | Local | Base | Novel | HM |
|---|---|---|---|---|
| Vision Only | 95.81 | 71.19 | 76.07 | 79.31 |
| Text Only | 95.99 | 71.19 | 76.13 | 79.38 |
| Both Vision & Text | 95.99 | 71.24 | 76.10 | 79.39 |
| pFedMMA (Ours) | **96.14** | **71.78** | **76.17** | **79.70** |

small but consistent differences in base, novel, and HM: aggregating text-only or both modalities yields a slight edge in HM over aggregating vision-only. Our full pFedMMA, which uses a single multi-modal shared projection rather than two separate modality-specific ones, further improves local, base, novel, and HM over all three variants, suggesting that tying the modalities through a unified

shared adapter provides a slightly stronger and more stable information-sharing mechanism without harming personalization.

## 4.4 LEARNING CURVES

To further analyze the convergence behavior of pFedMMA, we plot the average local accuracy over communication rounds across five different shot settings in Fig. 4. As shown, pFedMMA consistently attains high accuracy and converges faster than the baselines. Detailed results are provided in Fig. 5 in the Appendix.

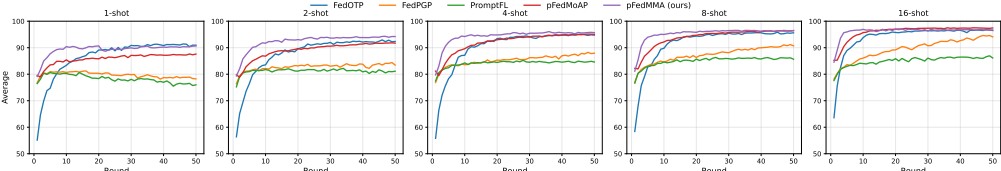

Figure 4: Accuracy learning curves of pFedMMA and baselines.

## 4.5 TRAINING COST ANALYSIS

Table 7 summarizes the computational and communication costs together with accuracy. PromptFL has the smallest footprint (8,192 trainable; 8,192/8,192 per round) but—most importantly—shows a marked drop in local accuracy (88.93%), indicating weaker personalization under heterogeneity. FedPGP increases local trainables to 12,416 without extra communication but incurs the highest memory (13,374 MiB) and slower training, and its HM accuracy (79.09%) lags its strong local accuracy (95.38%). FedOTP doubles prompt capacity (16,384 trainables; same 8,192/8,192 communication) and attains very high local accuracy (97.34%) but suffers extremely low HM accuracy (31.08%), suggesting poor cross-client generalization. pFedMoAP adds a local gating module, raising local trainables (74,240) and the per-round download (73,728) while achieving the shortest training time (902 s) and strong local accuracy (97.89%). pFedMMA (ours) communicates only the shared adapter blocks (3,072/3,072) while keeping 248,832 parameters local, yielding the best HM accuracy (84.15%) and competitive local accuracy (97.17%), thus offering the most favorable accuracy–communication trade-off. Please refer to Appendix E for additional details.

Table 7: Comparison of computation, communication, and accuracy for five personalized federated learning methods under CLIP ViT-B/16.

| Methods | # Local Trainable Param. | # Per-round Com. Param. (up/down) | Train Time (s) | GPU Mem. (MiB) | Avg. Local Acc. | Avg. HM Acc. |
|---|---|---|---|---|---|---|
| PromptFL Guo et al. (2023b) | 8,192 | 8,192 / 8,192 | 1,645 | 5,116 | 88.93 | 83.09 |
| FedPGP Cui et al. (2024) | 12,416 | 8,192 / 8,192 | 3,980 | 13,374 | 95.38 | 79.09 |
| FedOTP Li et al. (2024) | 16,384 | 8,192 / 8,192 | 1,328 | 3,014 | 97.34 | 31.08 |
| pFedMoAP Luo et al. (2025) | 74,240 | 8,192 / 73,728 | 902 | 3,108 | 97.89 | 71.05 |
| pFedMMA (Ours) | 248,832 | 3,072 / 3,072 | 2,175 | 4,634 | 97.17 | 84.15 |

## 5 CONCLUSION

In this work, we introduced pFedMMA, a novel personalized federated learning framework that leverages multi-modal adapters to adapt large-scale vision-language models under heterogeneous data conditions. The proposed architecture separates each adapter into modality-specific and shared projection components. Clients update all components locally, but only the shared projection is aggregated globally. This asymmetric optimization strategy enables client-specific adaptation while maintaining a globally aligned feature space for effective generalization. Moreover, the communication-efficient nature of the framework makes it scalable to real-world federated deployments. Our extensive experiments across diverse datasets demonstrate that pFedMMA consistently outperforms existing prompt-based PFL methods in both domain- and category-level generalization, while retaining strong personalization capabilities. This work can motivate further exploration of adapter-based architectures for personalized federated learning in multi-modal settings.

## ACKNOWLEDGMENTS

This work was supported by the National Science Foundation under Grant 2419982, Grant 2342253, and Grant 2236483.

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

## A   Pipelines of the Proposed Algorithm

For a better understanding of the steps of the designed algorithm, we present pFedMMA in Algorithm 1.

---

**Algorithm 1** pFedMMA

---

1: **Input:** Step size $\eta$, number of communication rounds $T$, number of local epoch $E$, number of clients $N$.
2: **for** communication round $t \leftarrow 1$ to $T$ **do**
3:     Select a subset of $|\mathcal{S}_t|$ clients, $\mathcal{S}_t$
4:     Send $\{\boldsymbol{W}_{\ell s}^t, \cdots, \boldsymbol{W}_{Ls}^t\}$ to the selected clients
5:     **for** clients $i \in \mathcal{S}_t$ in parallel **do**
6:         **for** local update $e \leftarrow 1$ to $E$ **do**
7:             for trainable parameters $\boldsymbol{W} \in \left\{ \boldsymbol{W}_{jd,i}^{(I)}, \boldsymbol{W}_{ju,i}^{(I)}, \boldsymbol{W}_{jd,i}^{(T)}, \boldsymbol{W}_{ju,i}^{(T)}, \boldsymbol{W}_{js,i} \right\}, \quad j \in \{\ell, \cdots, L\} :$
8:                 $\boldsymbol{W}_i^{t,e} = \boldsymbol{W}_i^{t,e-1} - \eta \nabla \mathcal{L}_{ce}(\boldsymbol{W}_i^{t,e-1})$
9:         **end for**
10:         Client $i$ sends $\boldsymbol{W}_{js,i}^{t,E}$ to the server
11:     **end for**
12:     At server: $\boldsymbol{W}_{js}^{t+1} = \sum_{i=1}^N p_i \boldsymbol{W}_{js,i}^{t,E}, \quad j \in \{\ell, \cdots, L\}$
13: **end for**
14: **Return:** $\left\{ \boldsymbol{W}_{jd,i}^{(I)}, \boldsymbol{W}_{ju,i}^{(I)}, \boldsymbol{W}_{jd,i}^{(T)}, \boldsymbol{W}_{ju,i}^{(T)}, \boldsymbol{W}_{js,i} \right\}, \quad i \in \{\ell, \cdots, L\}, \quad j \in \{\ell, \cdots, L\}$

---

## B   Related Work

### B.1   Personalized Federated Learning

Personalized Federated Learning (PFL) has emerged as a pivotal research direction to address the limitations of conventional federated learning McMahan et al. (2017) when faced with heterogeneous client data. Unlike standard FL, which learns a single global model, PFL aims to produce tailored models for individual clients, thus better coping with statistical and systemic heterogeneity Tan et al. (2022); Kulkarni et al. (2020). Several personalization strategies have been proposed, including local fine-tuning Mansour et al. (2020); Tan et al. (2022); Wang et al. (2019), regularization-based optimization Li et al. (2020; 2021b); T Dinh et al. (2020), and parameter decomposition into shared and client-specific components Arivazhagan et al. (2019); Oh et al.; Collins et al. (2021). Other methods pursue clustering of clients to exploit latent similarities Huang et al. (2021); Zhang et al.; Sattler et al. (2020); Ziad et al. (2024), or leverage attention mechanisms and adaptive layers Liang et al. (2020); Li et al. (2023a); Sun et al. (2023). To further improve adaptability, techniques like FedBN Li et al. and PartialFed Sun et al. (2021) address feature shift via local normalization or selective personalization. Hybrid global-local learning approaches have also been developed Deng et al. (2020); Chen & Chao (2022). FedOT Farnia et al. (2022) proposes learning optimal transport maps that align local distributions to a shared probability space, enabling a global classifier to be trained more effectively; personalization is achieved by composing this shared model with each client's transport map. While these approaches have demonstrated success, they typically center on traditional ML architectures and do not yet fully leverage the potential of large pre-trained models, such as vision-language or foundation models, for personalization.

### B.2   Federated Prompt Learning for VLMs

Federated Prompt Learning (FPL) extends the flexibility of prompt tuning for adapting large pre-trained models such as CLIP Radford et al. (2021) to the FL settings, enabling efficient and personalized downstream task adaptation across decentralized clients. Early works like CoOp Zhou et al. (2022b) and CoCoOp Zhou et al. (2022a) laid the foundation by introducing learnable continuous prompt vectors, which sparked interest in federated extensions. PromptFL Guo et al. (2023b) and FedPrompt Zhao et al. (2022) introduced FL-style prompt aggregation, performing FedAvg McMahan et al. (2017) over client-specific prompt updates. FedPR Feng et al. (2023) explores visual prompt learning within the null space of global prompts for MRI reconstruction, while FedAPT Su et al. (2022) focuses on domain-adaptive prompt tuning for cross-domain image classification. To en-

hance personalization, pFedPrompt Guo et al. (2023a) introduces a non-parametric attention module over local few-shot memory, and pFedPG Yang et al. (2023) and FedTPG Qiu et al. design server-side prompt generators to issue personalized prompts to each client. FedCLIP Lu et al. integrates attention-based adapters to better exploit the pre-trained model's knowledge. Furthermore, FedOTP Li et al. (2024) leverages Optimal Transport to align global and local prompts, and FedPGP **?** utilizes prompt-wise contrastive losses to better capture diverse category-level traits across clients. Recently, pFedMoAP Luo et al. (2025) rethinks prompt sharing by treating pre-aggregated prompts from other clients as non-local experts in a Mixture-of-Experts framework, enabling effective personalization via a lightweight, attention-based gating mechanism. Theoretical analysis of FPL Pan et al. (2024) provides deeper understanding of its convergence properties.

### B.3    Efficient Transfer Learning for VLMs

Traditional transfer learning approaches typically fine-tune all parameters of pre-trained VLMs Devlin et al. (2019); He et al. (2016), but this becomes increasingly impractical as model sizes scale up, especially under computational or data constraints. To mitigate this, the community has embraced parameter-efficient transfer learning strategies that modify only a small fraction of model parameters. Among these, prompt learning techniques, briefly introduced in the previous section, optimize lightweight vectors or tokens to steer the model without altering its backbone Zhou et al. (2022b;a); Lu et al. (2022); Khattak et al. (2023b). Although effective, they are often limited in their expressiveness or modality interaction. As a result, adapter-based methods have emerged as a powerful alternative. CLIP-Adapter Gao et al. (2024) and Tip-Adapter Zhang et al. (2022) inject lightweight MLP layers after the image encoder to refine visual representations. Tip-Adapter further improves efficiency by caching training features for fast inference. However, these image-only approaches neglect the cross-modal nature of VLMs. To address this, MMA Yang et al. (2024) introduces a multi-modal adapter architecture that fuses features across the vision and language branches via a shared representation space, enabling gradient flow between modalities. Similarly, other works explore deeper adapter integration, such as inserting adapters within self-attention and MLP blocks Jiang et al. (2022), allowing more granular control over the representation learning process. These advances mark a shift from single-stream to multi-stream adaptation, aligning with the unique demands of multi-modal tasks. In federated settings, where full model updates are prohibitive, adapter-based techniques offer a compelling balance between personalization, generalization, and communication efficiency—making them well-suited foundations for multi-model federated frameworks like ours.

### B.4    Federated Out-of-Distribution and Domain Generalization

A complementary line of work studies federated domain generalization (FedDG), where the goal is to train a single global model that generalizes to unseen target domains under client heterogeneity. FedDAT Chen et al. (2024) tackles multi-modal heterogeneous FL for foundation vision-language models via a dual-adapter teacher and mutual knowledge distillation, improving global performance across diverse vision-language tasks under domain shift. PLAN Gong et al. (2024) introduces a FedDG framework for pre-trained vision-language models based on visual and textual prompt learning and attention-based prompt aggregation, explicitly using a leave-one-domain-out protocol to adapt a global CLIP-style model to unseen domains. Other recent methods similarly design adapter- or prompt-based FedDG algorithms to enhance out-of-domain robustness of federated foundation models Li et al. (2023b); Yang et al. (2025); Lu et al. (2023). In contrast, our work is formulated in the *personalized federated learning* (PFL) setting, where each client maintains its own model (or adapter) and we explicitly optimize the personalization–generalization trade-off; consequently, we treat FedDG methods as complementary rather than direct baselines, and instead compare against personalized prompt- and adapter-based methods, while showing that our approach achieves comparable personalized performance and substantially stronger cross-domain generalization within this PFL protocol.

## C   Experimental Details

### C.1   Dataset Setup

For evaluation, we consider a broad set of eleven visual recognition benchmarks that span diverse tasks and levels of granularity. Table 8 provides a comprehensive overview, detailing the task type, number of classes, training and testing sizes, client splits, and the heterogeneity assumption used in our experiments.

The pathological partition setting is adopted for datasets such as Caltech101, Flowers102, OxfordPets, Food101, DTD, SUN397, and UCF101, where each client is assigned data corresponding to a limited number of classes, creating strong non-IID conditions. This simulates realistic personalization scenarios for fine-grained recognition, texture classification, scene recognition, and video action recognition.

For CIFAR-10 and CIFAR-100, we follow the common Dirichlet partitioning scheme with varying $\beta$ values to control the label skew among 100 clients. This allows systematic evaluation under different degrees of heterogeneity.

To capture the challenges of multi-domain learning, we also include Office-Caltech10 and DomainNet. Office-Caltech10 contains four domains (Amazon, Caltech, DSLR, Webcam), reflecting variations across acquisition devices and environments, while DomainNet consists of six domains (Clipart, Infograph, Painting, Quickdraw, Real, Sketch), which are significantly diverse and large-scale. For these benchmarks, we use 10 selected classes and evaluate both single-client-per-domain and multi-client-per-domain partitions.

Table 8: Statistical details of datasets used in experiments.

| Dataset | Task | #Classes | #Clients | Sample Rate | Training Size | Testing Size | Domains | Heterogeneity |
|---|---|---|---|---|---|---|---|---|
| Caltech101 Fei-Fei et al. (2004) | Object recognition | 100 | 10 | 100% | 4,128 | 2,465 | 1 | Pathological |
| Flowers102 Nilsback & Zisserman (2008) | Fine-grained flowers recognition | 102 | 10 | 100% | 4,093 | 2,463 | 1 | Pathological |
| OxfordPets Parkhi et al. (2012) | Fine-grained pets recognition | 37 | 10 | 100% | 2,944 | 3,669 | 1 | Pathological |
| Food101 Bossard et al. (2014) | Fine-grained food recognition | 101 | 10 | 100% | 50,500 | 30,300 | 1 | Pathological |
| DTD Cimpoi et al. (2014) | Texture recognition | 47 | 10 | 100% | 2,820 | 1,692 | 1 | Pathological |
| SUN397 Xiao et al. (2010) | Scene recognition | 397 | 10 | 100% | 76,128 | 21,750 | 1 | Pathological |
| UCF101 Soomro et al. (2012) | Action recognition (video) | 101 | 10 | 100% | 9,537 | 3,783 | 1 | Pathological |
| CIFAR-10 Krizhevsky et al. (2010) | Image classification | 10 | 100 | 10% | 50,000 | 10,000 | 1 | Dir($\beta$) |
| CIFAR-100 Krizhevsky et al. (2009) | Image classification | 100 | 100 | 10% | 50,000 | 10,000 | 1 | Dir($\beta$) |
| DomainNet Peng et al. (2019) | Image recognition | 10 | 4/8 | 100% | 18,278 | 4,573 | 6 | Dir($\beta$) |
| Office-Caltech10 Gong et al. (2012) | Image recognition | 10 | 6/12 | 100% | 2,025 | 508 | 4 | Dir($\beta$) |

### C.2   Experimental Setup

All models are trained using the SGD optimizer with a learning rate of $\eta = 0.001$. Each experiment is repeated three times with different random seeds, and we report the average performance. The final results are obtained by averaging performance across all clients. All experiments are implemented in PyTorch and run on NVIDIA A6000 GPUs.

**Base-to-Novel Class Generalization.** To evaluate generalization, we divide each dataset evenly into base and novel classes. Base classes are distributed across clients without overlap, such that each client only observes a subset during training. Clients train their local models on their own classes, and evaluation is performed on three levels: (i) local classes (the client's own training classes), (ii) base classes (classes seen by other clients but unseen locally), and (iii) novel classes (completely unseen during training). Accuracy is averaged across 10 clients.

**Feature & Label Shifts.** To evaluate robustness under realistic federated learning conditions, we conduct experiments with both label shift and feature shift using the DomainNet and Office-Caltech10 datasets. Each domain is partitioned into one or two clients using a Dirichlet distribution with varying $\beta$, resulting in 6 or 12 clients for DomainNet and 4 or 8 clients for Office-Caltech10. This setup generates heterogeneous client distributions, effectively simulating domain shifts.

**Personalization.** For personalization analysis, CIFAR-10 and CIFAR-100 are partitioned among 100 clients using a symmetric Dirichlet distribution. In addition, for the CLIP datasets, we follow the pathological partitioning strategy from the base-to-novel generalization setting, where classes are non-overlapping across 10 clients.

# D  ADDITIONAL EXPERIMENTS RESULTS

## D.1  BASE-TO-NOVEL CLASS GENERALIZATION

Table 9: Top-1 accuracy (%) of different methods across 7 datasets using ViT-B/16 as the backbone.

**Shots = 16**

| Method | Average on 7 datasets Local | Base | Novel | HM | SUN397 Local | Base | Novel | HM | Flowers102 Local | Base | Novel | HM | DTD Local | Base | Novel | HM |
|---|---|---|---|---|---|---|---|---|---|---|---|---|---|---|---|---|
| CLIP Radford et al. (2021) | 76.36 | 76.81 | 81.21 | 78.03 | 69.41 | 69.38 | 75.52 | 71.32 | 67.89 | 69.23 | 76.88 | 71.12 | 54.26 | 54.86 | 59.18 | 56.02 |
| PromptFL Guo et al. (2023b) | 88.93 | 88.95 | 75.36 | 83.09 | 77.73 | 77.71 | 72.96 | 76.07 | 97.37 | 97.06 | 63.62 | 82.66 | 80.23 | 80.21 | 45.29 | 63.81 |
| FedPGP Cui et al. (2024) | 95.38 | 76.49 | 71.68 | 79.09 | 94.29 | 54.88 | 57.76 | 65.02 | 99.67 | 72.44 | 58.65 | 73.37 | 89.03 | 71.03 | 50.94 | 66.75 |
| FedOTP Li et al. (2024) | 97.34 | 18.00 | 36.69 | 31.08 | 94.50 | 11.51 | 14.86 | 18.21 | 99.65 | 14.62 | 30.49 | 26.97 | 98.08 | 20.79 | 35.36 | 34.65 |
| pFedMoAP Luo et al. (2025) | 97.89 | 61.82 | 66.60 | 71.05 | 95.93 | 31.18 | 35.40 | 42.41 | 99.81 | 43.70 | 48.37 | 55.99 | 96.43 | 53.60 | 48.21 | 60.28 |
| pFedMMA (Ours) | 97.17 | 77.40 | 81.49 | 84.15 | 94.06 | 70.99 | 76.37 | 79.34 | 95.58 | 71.54 | 76.00 | 79.79 | 97.45 | 55.44 | 61.55 | 67.35 |
| Δ | −0.74% | +1.19% | +13.69% | +6.4% | −1.95% | +29.35% | +32.22% | +22.02% | −4.24% | −1.24% | +29.58% | +8.75% | −0.64% | −21.95% | +20.83% | +0.9% |

| Method | OxfordPets Local | Base | Novel | HM | Caltech101 Local | Base | Novel | HM | Food101 Local | Base | Novel | HM | UCF101 Local | Base | Novel | HM |
|---|---|---|---|---|---|---|---|---|---|---|---|---|---|---|---|---|
| CLIP Radford et al. (2021) | 89.45 | 89.42 | 96.81 | 91.77 | 96.14 | 97.22 | 94.21 | 95.84 | 89.40 | 89.42 | 90.70 | 89.84 | 68.00 | 68.15 | 75.18 | 70.29 |
| PromptFL Guo et al. (2023b) | 96.35 | 96.28 | 97.26 | 96.63 | 97.77 | 98.19 | 92.58 | 96.11 | 90.48 | 90.50 | 91.37 | 90.78 | 82.57 | 82.73 | 64.47 | 75.55 |
| FedPGP Cui et al. (2024) | 96.62 | 95.17 | 97.15 | 96.31 | 99.42 | 94.94 | 90.88 | 94.95 | 93.70 | 86.38 | 87.14 | 88.96 | 94.94 | 60.62 | 59.26 | 68.33 |
| FedOTP Li et al. (2024) | 100.00 | 11.60 | 51.22 | 25.92 | 99.94 | 36.47 | 62.77 | 56.23 | 95.69 | 17.29 | 37.97 | 31.70 | 93.54 | 13.62 | 24.19 | 23.91 |
| pFedMoAP Luo et al. (2025) | 99.92 | 77.61 | 92.05 | 88.87 | 99.92 | 94.07 | 92.43 | 95.37 | 97.49 | 69.86 | 83.51 | 82.09 | 95.79 | 62.74 | 66.23 | 72.33 |
| pFedMMA (Ours) | 100.00 | 88.50 | 96.60 | 94.78 | 100.00 | 96.53 | 94.29 | 96.88 | 97.45 | 89.15 | 90.77 | 92.32 | 95.63 | 69.61 | 74.88 | 78.58 |
| Δ | 0% | −7.01% | −0.57% | −1.59% | +0.06% | +1.67% | +2.01% | +1.58% | −0.04% | +3.21% | +4.17% | +3.78% | −0.17% | +10.95% | +13.06% | +8.64% |

**Shots = 8**

| Method | Average on 7 datasets Local | Base | Novel | HM | SUN397 Local | Base | Novel | HM | Flowers102 Local | Base | Novel | HM | DTD Local | Base | Novel | HM |
|---|---|---|---|---|---|---|---|---|---|---|---|---|---|---|---|---|
| CLIP Radford et al. (2021) | 76.36 | 76.81 | 81.21 | 78.03 | 69.41 | 69.38 | 75.52 | 71.32 | 67.89 | 69.23 | 76.88 | 71.12 | 54.26 | 54.86 | 59.18 | 56.02 |
| PromptFL Guo et al. (2023b) | 88.24 | 88.03 | 77.57 | 83.76 | 78.03 | 78.01 | 71.95 | 75.89 | 95.71 | 95.63 | 69.29 | 84.90 | 78.29 | 76.04 | 46.98 | 63.55 |
| FedPGP Cui et al. (2024) | 93.41 | 83.36 | 70.89 | 83.14 | 93.95 | 54.50 | 57.63 | 64.73 | 94.84 | 92.49 | 71.07 | 84.68 | 85.42 | 71.47 | 51.45 | 66.47 |
| FedOTP Li et al. (2024) | 96.63 | 24.30 | 42.92 | 38.30 | 93.41 | 11.90 | 18.00 | 19.96 | 99.73 | 20.47 | 45.03 | 37.00 | 96.99 | 23.33 | 42.48 | 39.11 |
| pFedMoAP Luo et al. (2025) | 97.04 | 71.31 | 71.73 | 77.58 | 95.16 | 45.80 | 49.43 | 57.06 | 99.75 | 66.88 | 61.99 | 72.98 | 94.44 | 58.11 | 52.10 | 63.84 |
| pFedMMA (Ours) | 96.66 | 79.29 | 81.61 | 84.28 | 92.71 | 70.90 | 76.25 | 78.94 | 95.52 | 72.31 | 76.43 | 80.25 | 95.32 | 56.30 | 61.57 | 67.42 |
| Δ | −0.39% | −4.88% | +13.77% | +1.37% | −2.57% | +30.09% | +32.31% | +21.95% | −4.24% | −21.82% | +7.54% | −5.23% | −1.72% | −21.23% | +18.18% | +1.43% |

| Method | OxfordPets Local | Base | Novel | HM | Caltech101 Local | Base | Novel | HM | Food101 Local | Base | Novel | HM | UCF101 Local | Base | Novel | HM |
|---|---|---|---|---|---|---|---|---|---|---|---|---|---|---|---|---|
| CLIP Radford et al. (2021) | 89.45 | 89.42 | 96.81 | 91.77 | 96.14 | 97.22 | 94.21 | 95.84 | 89.40 | 89.42 | 90.70 | 89.84 | 68.00 | 68.15 | 75.18 | 70.29 |
| PromptFL Guo et al. (2023b) | 95.53 | 95.43 | 97.32 | 96.09 | 97.31 | 98.26 | 94.43 | 96.64 | 90.44 | 90.46 | 90.67 | 90.52 | 82.39 | 82.37 | 72.36 | 78.75 |
| FedPGP Cui et al. (2024) | 96.09 | 94.93 | 96.66 | 95.89 | 98.13 | 98.26 | 93.47 | 96.57 | 90.57 | 90.38 | 91.04 | 90.66 | 94.85 | 81.48 | 74.94 | 82.96 |
| FedOTP Li et al. (2024) | 100.00 | 15.67 | 55.72 | 32.69 | 99.89 | 58.68 | 73.35 | 73.74 | 95.08 | 24.92 | 40.33 | 39.77 | 91.28 | 15.12 | 25.52 | 25.80 |
| pFedMoAP Luo et al. (2025) | 99.90 | 76.13 | 91.76 | 89.00 | 99.56 | 96.42 | 93.02 | 96.26 | 96.76 | 81.45 | 87.90 | 84.53 | 93.68 | 72.88 | 66.39 | 76.03 |
| pFedMMA (Ours) | 99.95 | 89.35 | 96.64 | 95.10 | 99.85 | 96.99 | 94.30 | 96.99 | 97.15 | 89.24 | 90.73 | 92.25 | 96.09 | 69.94 | 75.33 | 78.99 |
| Δ | −0.05% | −5.88% | −0.02% | −0.82% | −0.04% | −1.29% | +0.89% | +0.43% | +0.4% | −1.26% | +1.75% | +1.77% | +1.31% | −14.16% | +0.52% | −4.79% |

**Shots = 4**

| Method | Average on 7 datasets Local | Base | Novel | HM | SUN397 Local | Base | Novel | HM | Flowers102 Local | Base | Novel | HM | DTD Local | Base | Novel | HM |
|---|---|---|---|---|---|---|---|---|---|---|---|---|---|---|---|---|
| CLIP Radford et al. (2021) | 76.36 | 76.81 | 81.21 | 78.03 | 69.41 | 69.38 | 75.52 | 71.32 | 67.89 | 69.23 | 76.88 | 71.12 | 54.26 | 54.86 | 59.18 | 56.02 |
| PromptFL Guo et al. (2023b) | 87.12 | 86.90 | 79.06 | 84.29 | 77.60 | 77.57 | 76.11 | 77.09 | 95.05 | 94.68 | 65.79 | 84.72 | 73.56 | 71.41 | 56.88 | 64.46 |
| FedPGP Cui et al. (2024) | 90.23 | 85.18 | 78.15 | 83.73 | 83.72 | 71.91 | 69.78 | 74.66 | 94.48 | 92.43 | 72.82 | 85.38 | 82.69 | 67.51 | 51.65 | 64.84 |
| FedOTP Li et al. (2024) | 95.89 | 30.70 | 45.68 | 44.81 | 91.88 | 17.98 | 27.17 | 29.04 | 98.79 | 19.00 | 33.61 | 32.43 | 95.28 | 25.12 | 41.23 | 40.24 |
| pFedMoAP Luo et al. (2025) | 95.89 | 73.47 | 73.72 | 79.18 | 92.49 | 59.62 | 61.08 | 68.25 | 99.62 | 66.14 | 62.01 | 72.67 | 92.41 | 51.78 | 49.90 | 59.79 |
| pFedMMA (Ours) | 96.09 | 77.97 | 81.62 | 84.20 | 91.33 | 70.06 | 76.12 | 78.51 | 95.35 | 73.06 | 79.00 | 79.90 | 93.75 | 56.46 | 61.91 | 67.37 |
| Δ | +0.21% | −8.46% | +4.44% | +0.56% | −1.25% | −1.56% | +9.09% | +5.16% | −4.29% | −22.22% | +4.55% | −6.42% | −1.61% | −16.37% | +19.86% | +3.9% |

| Method | OxfordPets Local | Base | Novel | HM | Caltech101 Local | Base | Novel | HM | Food101 Local | Base | Novel | HM | UCF101 Local | Base | Novel | HM |
|---|---|---|---|---|---|---|---|---|---|---|---|---|---|---|---|---|
| CLIP Radford et al. (2021) | 89.45 | 89.42 | 96.81 | 91.77 | 96.14 | 97.22 | 94.21 | 95.84 | 89.40 | 89.42 | 90.70 | 89.84 | 68.00 | 68.15 | 75.18 | 70.29 |
| PromptFL Guo et al. (2023b) | 95.84 | 95.96 | 97.76 | 96.51 | 97.17 | 98.00 | 93.34 | 96.13 | 89.93 | 89.95 | 90.14 | 90.04 | 80.67 | 80.71 | 75.99 | 79.06 |
| FedPGP Cui et al. (2024) | 96.55 | 95.54 | 97.57 | 96.55 | 97.84 | 97.99 | 92.34 | 95.98 | 90.25 | 90.24 | 91.17 | 90.55 | 83.09 | 80.64 | 71.73 | 78.17 |
| FedOTP Li et al. (2024) | 99.95 | 41.83 | 70.13 | 62.28 | 99.73 | 65.97 | 77.59 | 78.79 | 94.49 | 29.38 | 46.57 | 45.39 | 91.20 | 15.60 | 23.48 | 25.50 |
| pFedMoAP Luo et al. (2025) | 99.75 | 85.69 | 94.08 | 92.81 | 99.08 | 95.73 | 92.73 | 95.85 | 95.49 | 81.68 | 86.57 | 87.55 | 92.40 | 73.42 | 69.67 | 77.33 |
| pFedMMA (Ours) | 99.90 | 89.68 | 96.63 | 95.21 | 99.85 | 96.93 | 94.40 | 97.01 | 96.60 | 89.34 | 90.83 | 92.15 | 95.87 | 70.71 | 75.37 | 79.28 |
| Δ | −0.05% | −6.13% | −0.96% | −1.39% | +0.12% | −1.08% | +1.8% | +1.07% | +1.16% | −1% | −0.37% | +1.77% | +3.76% | −12.31% | +5.07% | +1.42% |

**Shots = 2**

| Method | Average on 7 datasets Local | Base | Novel | HM | SUN397 Local | Base | Novel | HM | Flowers102 Local | Base | Novel | HM | DTD Local | Base | Novel | HM |
|---|---|---|---|---|---|---|---|---|---|---|---|---|---|---|---|---|
| CLIP Radford et al. (2021) | 76.36 | 76.81 | 81.21 | 78.03 | 69.41 | 69.38 | 75.52 | 71.32 | 67.89 | 69.23 | 76.88 | 71.12 | 54.26 | 54.86 | 59.18 | 56.02 |
| PromptFL Guo et al. (2023b) | 85.23 | 85.02 | 77.30 | 81.97 | 75.66 | 75.65 | 75.64 | 75.65 | 92.71 | 92.69 | 65.25 | 81.30 | 67.69 | 65.28 | 48.43 | 59.12 |
| FedPGP Cui et al. (2024) | 86.24 | 83.88 | 78.02 | 82.56 | 76.42 | 74.97 | 72.34 | 74.54 | 90.98 | 89.38 | 68.33 | 81.16 | 73.66 | 62.56 | 56.85 | 63.63 |
| FedOTP Li et al. (2024) | 94.10 | 33.34 | 42.49 | 46.54 | 87.53 | 26.06 | 31.85 | 36.95 | 97.77 | 18.57 | 29.49 | 40.34 | 91.71 | 26.94 | 37.97 | 40.34 |
| pFedMoAP Luo et al. (2025) | 93.11 | 74.45 | 73.66 | 79.16 | 85.87 | 70.76 | 68.06 | 74.13 | 99.81 | 65.00 | 61.84 | 72.84 | 82.64 | 53.28 | 46.23 | 57.14 |
| pFedMMA (Ours) | 94.57 | 78.13 | 77.37 | 83.68 | 89.23 | 71.48 | 76.46 | 77.58 | 94.42 | 72.67 | 75.98 | 79.16 | 89.72 | 56.48 | 61.92 | 66.67 |
| Δ | +0.5% | −6.86% | −0.83% | +1.36% | +1.94% | −4.66% | +5.7% | +4.08% | −4.44% | −18.75% | +11.22% | −2.46% | −2.17% | −9.72% | +8.92% | +4.78% |

| Method | OxfordPets Local | Base | Novel | HM | Caltech101 Local | Base | Novel | HM | Food101 Local | Base | Novel | HM | UCF101 Local | Base | Novel | HM |
|---|---|---|---|---|---|---|---|---|---|---|---|---|---|---|---|---|
| CLIP Radford et al. (2021) | 89.45 | 89.42 | 96.81 | 91.77 | 96.14 | 97.22 | 94.21 | 95.84 | 89.40 | 89.42 | 90.70 | 89.84 | 68.00 | 68.15 | 75.18 | 70.29 |
| PromptFL Guo et al. (2023b) | 95.03 | 94.90 | 96.92 | 95.61 | 96.86 | 97.61 | 93.34 | 95.90 | 89.93 | 89.73 | 89.72 | 89.72 | 78.92 | 79.06 | 71.82 | 76.45 |
| FedPGP Cui et al. (2024) | 94.88 | 94.55 | 97.27 | 95.55 | 97.06 | 97.10 | 92.99 | 95.68 | 89.47 | 89.46 | 90.48 | 89.80 | 81.22 | 79.16 | 72.88 | 77.59 |
| FedOTP Li et al. (2024) | 100.00 | 38.63 | 55.59 | 55.68 | 99.64 | 75.35 | 81.09 | 84.18 | 93.16 | 27.49 | 44.35 | 43.07 | 88.87 | 20.34 | 17.13 | 25.25 |
| pFedMoAP Luo et al. (2025) | 99.69 | 85.76 | 90.07 | 91.48 | 99.33 | 96.07 | 91.76 | 95.62 | 94.97 | 81.21 | 86.27 | 87.12 | 90.46 | 69.10 | 71.21 | 75.82 |
| pFedMMA (Ours) | 99.64 | 89.46 | 96.80 | 95.10 | 99.40 | 97.07 | 94.14 | 96.95 | 95.59 | 89.40 | 90.76 | 91.78 | 90.90 | 70.36 | 75.12 | 78.52 |
| Δ | −0.36% | −5.38% | −0.48% | −0.47% | +0.16% | −0.03% | +1.24% | +1.33% | +0.65% | −0.07% | +0.31% | +2.2% | +3.47% | −11.12% | +3.07% | +1.2% |

**Shots = 1**

| Method | Average on 7 datasets Local | Base | Novel | HM | SUN397 Local | Base | Novel | HM | Flowers102 Local | Base | Novel | HM | DTD Local | Base | Novel | HM |
|---|---|---|---|---|---|---|---|---|---|---|---|---|---|---|---|---|
| CLIP Radford et al. (2021) | 76.36 | 76.81 | 81.21 | 78.03 | 69.41 | 69.38 | 75.52 | 71.32 | 67.89 | 69.23 | 76.88 | 71.12 | 54.26 | 54.86 | 59.18 | 56.02 |
| PromptFL Guo et al. (2023b) | 81.99 | 81.93 | 78.83 | 80.79 | 73.80 | 73.77 | 75.11 | 74.22 | 83.12 | 83.67 | 71.28 | 78.92 | 60.51 | 58.22 | 50.48 | 56.06 |
| FedPGP Cui et al. (2024) | 82.48 | 82.06 | 78.08 | 80.83 | 74.36 | 74.07 | 76.63 | 75.00 | 84.18 | 82.69 | 67.23 | 77.23 | 61.34 | 60.01 | 50.91 | 57.02 |
| FedOTP Li et al. (2024) | 93.51 | 32.72 | 47.55 | 47.24 | 86.48 | 27.51 | 34.68 | 39.09 | 95.88 | 22.83 | 45.91 | 44.05 | 88.61 | 32.44 | 41.28 | 45.22 |
| pFedMoAP Luo et al. (2025) | 87.65 | 75.82 | 76.35 | 79.30 | 81.90 | 68.22 | 68.14 | 72.21 | 82.70 | 70.37 | 76.55 | 75.69 | 75.56 | 55.00 | 50.71 | 58.67 |
| pFedMMA (Ours) | 92.40 | 71.66 | 80.77 | 83.50 | 84.42 | 72.56 | 76.30 | 78.50 | 92.76 | 70.37 | 76.55 | 79.23 | 86.67 | 56.31 | 61.77 | 65.77 |
| Δ | −1.19% | −5.36% | +3.64% | +3.3% | −2.38% | −4.25% | −0.33% | +4.67% | −3.25% | −15.9% | +7.39% | +0.39% | −2.19% | −6.17% | +20.41% | +15.35% |

| Method | OxfordPets Local | Base | Novel | HM | Caltech101 Local | Base | Novel | HM | Food101 Local | Base | Novel | HM | UCF101 Local | Base | Novel | HM |
|---|---|---|---|---|---|---|---|---|---|---|---|---|---|---|---|---|
| CLIP Radford et al. (2021) | 89.45 | 89.42 | 96.81 | 91.77 | 96.14 | 97.22 | 94.21 | 95.84 | 89.40 | 89.42 | 90.70 | 89.84 | 68.00 | 68.15 | 75.18 | 70.29 |
| PromptFL Guo et al. (2023b) | 94.89 | 94.95 | 97.43 | 95.74 | 96.11 | 97.29 | 94.43 | 95.93 | 89.45 | 89.46 | 90.02 | 89.64 | 76.05 | 76.16 | 73.07 | 75.07 |
| FedPGP Cui et al. (2024) | 94.88 | 94.75 | 96.66 | 95.42 | 96.30 | 97.42 | 92.96 | 95.52 | 89.04 | 89.04 | 90.91 | 89.62 | 77.26 | 76.47 | 74.23 | 75.96 |
| FedOTP Li et al. (2024) | 99.95 | 23.38 | 65.26 | 44.05 | 99.15 | 60.79 | 73.46 | 74.72 | 95.88 | 40.98 | 50.67 | 54.98 | 88.61 | 21.10 | 21.59 | 28.57 |
| pFedMoAP Luo et al. (2025) | 98.36 | 88.83 | 94.92 | 93.87 | 98.40 | 95.63 | 92.29 | 95.37 | 92.62 | 83.06 | 87.32 | 87.49 | 90.46 | 69.62 | 64.49 | 71.82 |
| pFedMMA (Ours) | 98.83 | 89.64 | 97.02 | 94.99 | 99.36 | 97.04 | 94.36 | 96.88 | 93.74 | 89.44 | 90.83 | 91.31 | 91.03 | 69.91 | 75.47 | 77.84 |
| Δ | −1.12% | −5.39% | −0.42% | −0.78% | +0.21% | −0.39% | −0.07% | +0.99% | −2.23% | −0.02% | −0.09% | +1.68% | +0.63% | −8.58% | +1.67% | +2.47% |

Table 10: Top-1 accuracy (%) of different methods across 7 datasets using ViT-B/32 as the backbone.

**Shots = 16**

| Method | OxfordPets Local | Base | Novel | HM | SUN397 Local | Base | Novel | HM | DTD Local | Base | Novel | HM |
|---|---|---|---|---|---|---|---|---|---|---|---|---|
| CLIP Radford et al. (2021) | 86.40 | 86.87 | 96.09 | 89.57 | 69.81 | 69.78 | 72.99 | 70.83 | 52.27 | 53.13 | 54.47 | 53.27 |
| PromptFL Guo et al. (2023b) | 93.78 | 93.83 | 96.53 | 94.70 | 75.38 | 75.35 | 68.04 | 72.75 | 77.22 | 76.85 | 57.00 | 68.96 |
| FedPGP Cui et al. (2024) | 95.01 | 93.01 | 94.31 | 94.10 | 95.33 | 38.76 | 47.21 | 52.20 | 88.49 | 61.99 | 41.74 | 58.38 |
| FedOTP Li et al. (2024) | 99.90 | 10.19 | 32.80 | 21.64 | 90.06 | 9.03 | 5.79 | 10.18 | 97.70 | 10.68 | 18.48 | 18.99 |
| pFedMoAP Luo et al. (2025) | 100.00 | 30.45 | 63.43 | 51.19 | 94.96 | 24.98 | 26.71 | 34.09 | 96.66 | 16.09 | 24.70 | 26.55 |
| pFedMMA (Ours) | 94.05 | 87.92 | 95.74 | 92.45 | 92.09 | 70.65 | 73.77 | 77.78 | 72.31 | 54.62 | 53.93 | 59.19 |
| Δ | | | | −2.38% | | | | +6.91% | | | | −14.17% |

| Method | Caltech101 Local | Base | Novel | HM | Food101 Local | Base | Novel | HM | UCF101 Local | Base | Novel | HM |
|---|---|---|---|---|---|---|---|---|---|---|---|---|
| CLIP Radford et al. (2021) | 92.63 | 94.00 | 94.00 | 93.54 | 84.33 | 84.27 | 85.43 | 84.67 | 65.16 | 65.36 | 71.28 | 67.15 |
| PromptFL Guo et al. (2023b) | 95.85 | 97.29 | 93.01 | 95.35 | 86.29 | 86.31 | 86.03 | 86.21 | 79.95 | 79.89 | 60.30 | 72.10 |
| FedPGP Cui et al. (2024) | 98.28 | 93.63 | 88.50 | 93.30 | 89.65 | 83.55 | 85.85 | 86.28 | 94.64 | 53.80 | 55.73 | 63.70 |
| FedOTP Li et al. (2024) | 99.38 | 11.46 | 23.17 | 21.36 | 92.21 | 9.63 | 14.03 | 16.13 | 88.36 | 8.84 | 6.15 | 10.45 |
| pFedMoAP Luo et al. (2025) | 99.92 | 51.79 | 65.33 | 67.23 | 96.41 | 31.56 | 53.51 | 49.39 | 95.28 | 24.08 | 31.24 | 35.70 |
| pFedMMA (Ours) | 98.56 | 94.74 | 93.94 | 95.70 | 95.31 | 84.59 | 85.59 | 88.24 | 87.80 | 66.91 | 71.53 | 74.41 |
| Δ | | | | +0.37% | | | | +2.27% | | | | +3.2% |

**Shots = 8**

| Method | OxfordPets Local | Base | Novel | HM | SUN397 Local | Base | Novel | HM | DTD Local | Base | Novel | HM |
|---|---|---|---|---|---|---|---|---|---|---|---|---|
| CLIP Radford et al. (2021) | 86.40 | 86.87 | 96.09 | 89.57 | 69.81 | 69.78 | 72.99 | 70.83 | 52.27 | 53.13 | 54.47 | 53.27 |
| PromptFL Guo et al. (2023b) | 93.34 | 93.57 | 96.03 | 94.30 | 75.93 | 75.91 | 73.61 | 75.13 | 75.05 | 72.34 | 49.64 | 63.43 |
| FedPGP Cui et al. (2024) | 94.87 | 91.69 | 92.94 | 93.15 | 93.15 | 55.78 | 57.24 | 65.03 | 78.66 | 72.76 | 45.61 | 62.01 |
| FedOTP Li et al. (2024) | 99.85 | 11.08 | 37.61 | 23.65 | 87.18 | 8.78 | 4.64 | 8.80 | 95.84 | 10.11 | 19.37 | 18.64 |
| pFedMoAP Luo et al. (2025) | 100.00 | 33.19 | 63.15 | 53.61 | 94.79 | 26.54 | 23.41 | 32.99 | 95.58 | 19.78 | 32.68 | 32.74 |
| pFedMMA (Ours) | 93.80 | 88.65 | 95.86 | 92.67 | 89.77 | 70.19 | 73.41 | 76.91 | 68.01 | 55.68 | 54.34 | 58.75 |
| Δ | | | | −1.73% | | | | +2.37% | | | | −7.38% |

| Method | Caltech101 Local | Base | Novel | HM | Food101 Local | Base | Novel | HM | UCF101 Local | Base | Novel | HM |
|---|---|---|---|---|---|---|---|---|---|---|---|---|
| CLIP Radford et al. (2021) | 92.63 | 94.00 | 94.00 | 93.54 | 84.33 | 84.27 | 85.43 | 84.67 | 65.16 | 65.36 | 71.28 | 67.15 |
| PromptFL Guo et al. (2023b) | 96.89 | 97.93 | 92.25 | 95.63 | 86.16 | 86.18 | 87.63 | 86.65 | 80.58 | 80.25 | 61.93 | 73.14 |
| FedPGP Cui et al. (2024) | 96.46 | 97.09 | 91.32 | 94.89 | 86.28 | 86.27 | 87.96 | 86.83 | 85.02 | 78.41 | 66.56 | 75.87 |
| FedOTP Li et al. (2024) | 99.03 | 11.99 | 27.97 | 23.21 | 89.94 | 9.44 | 18.90 | 17.65 | 82.82 | 8.31 | 7.84 | 11.54 |
| pFedMoAP Luo et al. (2025) | 99.72 | 80.74 | 80.58 | 86.15 | 96.21 | 54.37 | 68.67 | 69.21 | 94.98 | 40.58 | 37.06 | 48.27 |
| pFedMMA (Ours) | 97.87 | 94.88 | 93.64 | 95.43 | 95.17 | 84.73 | 85.72 | 88.30 | 83.06 | 66.59 | 71.07 | 72.94 |
| Δ | | | | −0.21% | | | | +1.69% | | | | −3.86% |

**Shots = 4**

| Method | OxfordPets Local | Base | Novel | HM | SUN397 Local | Base | Novel | HM | DTD Local | Base | Novel | HM |
|---|---|---|---|---|---|---|---|---|---|---|---|---|
| CLIP Radford et al. (2021) | 86.40 | 86.87 | 96.09 | 89.57 | 69.81 | 69.78 | 72.99 | 70.83 | 52.27 | 53.13 | 54.47 | 53.27 |
| PromptFL Guo et al. (2023b) | 93.73 | 93.99 | 96.87 | 94.84 | 75.53 | 75.50 | 72.24 | 74.39 | 71.16 | 68.87 | 46.38 | 59.84 |
| FedPGP Cui et al. (2024) | 93.93 | 93.23 | 91.26 | 92.79 | 83.31 | 70.90 | 66.64 | 72.97 | 74.70 | 68.33 | 44.37 | 59.34 |
| FedOTP Li et al. (2024) | 99.80 | 10.21 | 31.68 | 21.50 | 81.32 | 8.38 | 7.93 | 11.64 | 95.72 | 10.85 | 22.77 | 20.47 |
| pFedMoAP Luo et al. (2025) | 100.00 | 33.13 | 65.34 | 54.07 | 93.34 | 38.49 | 37.64 | 47.42 | 94.77 | 16.43 | 31.68 | 29.13 |
| pFedMMA (Ours) | 93.64 | 88.23 | 96.33 | 92.61 | 83.44 | 70.92 | 73.66 | 75.64 | 67.64 | 54.93 | 55.06 | 58.65 |
| Δ | | | | −2.35% | | | | +1.68% | | | | −1.99% |

| Method | Caltech101 Local | Base | Novel | HM | Food101 Local | Base | Novel | HM | UCF101 Local | Base | Novel | HM |
|---|---|---|---|---|---|---|---|---|---|---|---|---|
| CLIP Radford et al. (2021) | 92.63 | 94.00 | 94.00 | 93.54 | 84.33 | 84.27 | 85.43 | 84.67 | 65.16 | 65.36 | 71.28 | 67.15 |
| PromptFL Guo et al. (2023b) | 95.44 | 96.90 | 93.12 | 95.13 | 85.59 | 85.61 | 87.55 | 86.24 | 78.00 | 77.82 | 64.20 | 72.73 |
| FedPGP Cui et al. (2024) | 95.50 | 96.48 | 90.39 | 94.05 | 86.11 | 86.09 | 88.01 | 86.73 | 81.62 | 76.40 | 66.31 | 74.22 |
| FedOTP Li et al. (2024) | 99.08 | 18.93 | 39.24 | 33.94 | 89.84 | 9.39 | 18.42 | 17.45 | 83.32 | 8.34 | 7.10 | 11.00 |
| pFedMoAP Luo et al. (2025) | 99.77 | 81.21 | 82.39 | 87.00 | 95.75 | 52.96 | 65.93 | 67.43 | 95.52 | 42.63 | 43.05 | 52.49 |
| pFedMMA (Ours) | 97.32 | 94.80 | 93.84 | 95.30 | 94.33 | 84.15 | 85.33 | 87.71 | 84.53 | 68.04 | 71.68 | 74.11 |
| Δ | | | | +0.18% | | | | +1.13% | | | | −0.15% |

**Shots = 2**

| Method | OxfordPets Local | Base | Novel | HM | SUN397 Local | Base | Novel | HM | DTD Local | Base | Novel | HM |
|---|---|---|---|---|---|---|---|---|---|---|---|---|
| CLIP Radford et al. (2021) | 86.40 | 86.87 | 96.09 | 89.57 | 69.81 | 69.78 | 72.99 | 70.83 | 52.27 | 53.13 | 54.47 | 53.27 |
| PromptFL Guo et al. (2023b) | 90.62 | 91.44 | 96.87 | 92.90 | 74.08 | 74.06 | 73.81 | 73.98 | 61.20 | 60.88 | 47.95 | 55.95 |
| FedPGP Cui et al. (2024) | 92.05 | 92.41 | 91.80 | 92.09 | 74.78 | 74.36 | 73.70 | 74.28 | 67.13 | 64.93 | 54.77 | 61.78 |
| FedOTP Li et al. (2024) | 99.43 | 13.86 | 48.24 | 29.14 | 71.89 | 8.61 | 8.60 | 12.18 | 92.27 | 10.31 | 22.19 | 19.62 |
| pFedMoAP Luo et al. (2025) | 100.00 | 28.97 | 67.42 | 50.55 | 89.80 | 54.74 | 57.58 | 64.14 | 92.25 | 21.92 | 32.49 | 34.39 |
| pFedMMA (Ours) | 93.18 | 88.30 | 95.73 | 92.30 | 79.27 | 71.52 | 74.10 | 74.83 | 62.18 | 55.67 | 54.84 | 57.38 |
| Δ | | | | −0.65% | | | | +0.74% | | | | −7.12% |

| Method | Caltech101 Local | Base | Novel | HM | Food101 Local | Base | Novel | HM | UCF101 Local | Base | Novel | HM |
|---|---|---|---|---|---|---|---|---|---|---|---|---|
| CLIP Radford et al. (2021) | 92.63 | 94.00 | 94.00 | 93.54 | 84.33 | 84.27 | 85.43 | 84.67 | 65.16 | 65.36 | 71.28 | 67.15 |
| PromptFL Guo et al. (2023b) | 95.27 | 96.45 | 91.59 | 94.39 | 85.09 | 85.10 | 86.97 | 85.71 | 75.88 | 75.85 | 64.31 | 71.58 |
| FedPGP Cui et al. (2024) | 94.62 | 95.53 | 90.92 | 93.65 | 85.95 | 85.94 | 87.81 | 86.56 | 79.23 | 75.44 | 63.50 | 72.07 |
| FedOTP Li et al. (2024) | 97.95 | 14.81 | 32.70 | 27.70 | 88.58 | 10.78 | 18.58 | 19.00 | 82.00 | 9.36 | 8.39 | 12.59 |
| pFedMoAP Luo et al. (2025) | 99.62 | 81.01 | 80.41 | 86.16 | 95.01 | 52.55 | 68.69 | 68.00 | 93.63 | 43.63 | 43.39 | 52.96 |
| pFedMMA (Ours) | 97.89 | 94.70 | 93.73 | 95.41 | 92.87 | 84.98 | 85.87 | 87.77 | 81.02 | 67.09 | 71.78 | 72.85 |
| Δ | | | | +1.08% | | | | +1.4% | | | | +1.08% |

**Shots = 1**

| Method | OxfordPets Local | Base | Novel | HM | SUN397 Local | Base | Novel | HM | DTD Local | Base | Novel | HM |
|---|---|---|---|---|---|---|---|---|---|---|---|---|
| CLIP Radford et al. (2021) | 86.40 | 86.87 | 96.09 | 89.57 | 69.81 | 69.78 | 72.99 | 70.83 | 52.27 | 53.13 | 54.47 | 53.27 |
| PromptFL Guo et al. (2023b) | 91.70 | 92.29 | 97.04 | 93.62 | 72.86 | 72.82 | 73.56 | 73.08 | 56.71 | 57.29 | 44.93 | 52.31 |
| FedPGP Cui et al. (2024) | 86.86 | 88.71 | 87.98 | 87.84 | 74.54 | 74.34 | 87.66 | 78.38 | 60.76 | 57.93 | 55.83 | 58.10 |
| FedOTP Li et al. (2024) | 99.39 | 11.81 | 34.94 | 24.32 | 70.75 | 7.69 | 6.75 | 10.26 | 89.47 | 13.66 | 20.19 | 22.40 |
| pFedMoAP Luo et al. (2025) | 100.00 | 26.32 | 52.30 | 44.70 | 85.99 | 52.37 | 54.12 | 60.97 | 89.77 | 23.96 | 31.14 | 35.30 |
| pFedMMA (Ours) | 91.55 | 88.09 | 96.39 | 91.88 | 75.86 | 71.02 | 73.87 | 73.53 | 59.72 | 55.31 | 54.89 | 56.56 |
| Δ | | | | −1.86% | | | | −6.19% | | | | −2.65% |

| Method | Caltech101 Local | Base | Novel | HM | Food101 Local | Base | Novel | HM | UCF101 Local | Base | Novel | HM |
|---|---|---|---|---|---|---|---|---|---|---|---|---|
| CLIP Radford et al. (2021) | 92.63 | 94.00 | 94.00 | 93.54 | 84.33 | 84.27 | 85.43 | 84.67 | 65.16 | 65.36 | 71.28 | 67.15 |
| PromptFL Guo et al. (2023b) | 94.35 | 96.26 | 92.58 | 94.37 | 85.68 | 85.71 | 87.91 | 86.42 | 73.17 | 73.16 | 69.17 | 71.78 |
| FedPGP Cui et al. (2024) | 93.66 | 95.34 | 91.74 | 93.56 | 85.50 | 85.52 | 87.66 | 86.21 | 74.54 | 74.24 | 70.64 | 73.10 |
| FedOTP Li et al. (2024) | 97.88 | 14.77 | 30.07 | 26.98 | 85.38 | 11.43 | 22.85 | 20.98 | 78.14 | 7.86 | 9.14 | 12.03 |
| pFedMoAP Luo et al. (2025) | 99.14 | 77.97 | 77.75 | 83.86 | 93.62 | 56.89 | 68.39 | 69.96 | 90.57 | 35.69 | 40.75 | 47.17 |
| pFedMMA (Ours) | 94.77 | 94.18 | 93.83 | 94.26 | 91.67 | 84.78 | 85.84 | 87.33 | 79.28 | 67.30 | 71.29 | 72.29 |
| Δ | | | | −0.12% | | | | +1.05% | | | | −1.11% |

## D.2 MODEL EVALUATION ON FEATURE & LABEL SHIFTS

Table 11: Average test accuracy (%) of different methods on DomainNet and Office-Caltech10 with lable shift and domain shift using Dirichlet partitioning.

| Dataset | Office | | | | | | DomainNet | | | | | |
|---|---|---|---|---|---|---|---|---|---|---|---|---|
| #$\beta$ | 0.1 | 0.3 | 0.5 | 1 | 5 | 10 | 0.1 | 0.3 | 0.5 | 1 | 5 | 10 |
| **One domain for one client** | | | | | | | | | | | | |
| CLIP Radford et al. (2021) | 8.24 | 7.78 | 9.60 | 8.98 | 8.98 | 9.56 | 10.27 | 10.15 | 10.11 | 9.79 | 10.37 | 10.52 |
| PromptFL Guo et al. (2023b) | 14.53 | 15.39 | 15.61 | 14.32 | 15.57 | 14.36 | 12.52 | 11.77 | 11.81 | 12.21 | 11.66 | 11.77 |
| FedPGP Cui et al. (2024) | 14.18 | 16.88 | 14.17 | 12.39 | 16.13 | 13.07 | 14.55 | 13.55 | 14.15 | 14.29 | 14.18 | 14.34 |
| pFedMoAP Luo et al. (2025) | 12.65 | 16.14 | 12.27 | 14.19 | 14.70 | 17.03 | 14.14 | 13.89 | 14.30 | 14.14 | 14.38 | 13.55 |
| pFedMMA (Ours) | 21.08 | 22.38 | 19.06 | 20.43 | 18.42 | 18.73 | 36.18 | 37.06 | 42.55 | 43.31 | 46.13 | 34.69 |
| **One domain for two clients** | | | | | | | | | | | | |
| CLIP Radford et al. (2021) | 8.83 | 9.10 | 9.11 | 9.67 | 6.61 | 12.51 | 10.59 | 10.29 | 10.11 | 9.81 | 9.24 | 10.00 |
| PromptFL Guo et al. (2023b) | 15.99 | 15.29 | 16.34 | 14.85 | 16.14 | 14.43 | 11.83 | 12.58 | 11.24 | 11.27 | 11.57 | 11.55 |
| FedPGP Cui et al. (2024) | 22.55 | 19.29 | 20.71 | 21.96 | 19.63 | 15.19 | 26.08 | 26.30 | 24.90 | 21.22 | 16.14 | 15.07 |
| pFedMoAP Luo et al. (2025) | 22.73 | 23.06 | 19.55 | 21.67 | 16.57 | 19.02 | 24.99 | 24.79 | 24.65 | 21.59 | 16.43 | 15.24 |
| pFedMMA (Ours) | 21.66 | 22.07 | 21.33 | 18.47 | 20.96 | 17.73 | 49.45 | 37.61 | 47.17 | 48.95 | 46.90 | 48.54 |

Table 12: Test accuracy (%) of different methods on DomainNet and Office-Caltech10 with lable shift and domain shift using Dirichlet partitioning.

| Method | Office-Caltech10 | | | | | DomainNet | | | | | | |
|---|---|---|---|---|---|---|---|---|---|---|---|---|
| | Amazon | Caltech | DSLR | Webcam | Avg. | Clipart | Infograph | Painting | Quickdraw | Real | Sketch | Avg. |
| $\beta = 0.5$ | | | | | | | | | | | | |
| **One domain for one client** | | | | | | | | | | | | |
| CLIP | 10.42 | 5.33 | 12.50 | 10.17 | 9.60 | 8.37 | 10.50 | 12.12 | 10.40 | 8.79 | 10.47 | 10.11 |
| PromptFL | 8.85 | 23.73 | 11.11 | 18.75 | 15.61 | 12.16 | 12.33 | 11.79 | 9.40 | 13.25 | 11.91 | 11.81 |
| FedPGP | 11.98 | 9.78 | 28.13 | 6.78 | 14.17 | 14.07 | 19.18 | 14.05 | 10.30 | 13.39 | 13.90 | 14.15 |
| pFedMoAP | 5.21 | 9.33 | 12.50 | 22.03 | 12.27 | 13.69 | 19.18 | 15.35 | 11.50 | 12.90 | 13.18 | 14.30 |
| pFedMMA (Ours) | 10.71 | 17.81 | 17.46 | 30.26 | 19.06 | 50.38 | 48.82 | 0.00 | 32.56 | 37.31 | 86.21 | 42.55 |
| **One domain for two clients** | | | | | | | | | | | | |
| CLIP | 11.78 | 6.21 | 9.92 | 8.51 | 9.11 | 8.99 | 10.69 | 11.20 | 10.85 | 9.53 | 9.39 | 10.11 |
| PromptFL | 10.35 | 15.83 | 32.06 | 7.13 | 16.34 | 11.02 | 1.65 | 11.20 | 8.95 | 13.89 | 20.75 | 11.24 |
| FedPGP | 20.34 | 19.12 | 20.85 | 22.52 | 20.71 | 24.77 | 31.87 | 23.87 | 22.87 | 22.40 | 23.64 | 24.90 |
| pFedMoAP | 20.01 | 24.45 | 18.02 | 15.73 | 19.55 | 24.77 | 30.93 | 26.09 | 20.46 | 22.59 | 23.10 | 24.65 |
| pFedMMA (Ours) | 9.26 | 29.15 | 33.26 | 13.64 | 21.33 | 50.38 | 23.81 | 60.27 | 61.44 | 40.35 | 46.79 | 47.17 |
| $\beta = 0.3$ | | | | | | | | | | | | |
| **One domain for one client** | | | | | | | | | | | | |
| CLIP | 8.33 | 12.89 | 3.13 | 6.78 | 7.78 | 10.08 | 9.44 | 10.82 | 10.30 | 10.68 | 9.57 | 10.15 |
| PromptFL | 11.86 | 10.94 | 25.00 | 13.78 | 15.39 | 10.27 | 10.93 | 11.47 | 10.70 | 11.91 | 14.31 | 11.60 |
| FedPGP | 6.77 | 11.11 | 34.37 | 15.25 | 16.88 | 13.50 | 19.33 | 14.22 | 10.10 | 13.39 | 14.08 | 14.10 |
| pFedMoAP | 13.02 | 14.67 | 25.00 | 11.86 | 16.14 | 13.31 | 19.33 | 14.86 | 9.60 | 11.99 | 14.26 | 13.89 |
| pFedMMA (Ours) | 14.29 | 15.07 | 28.57 | 31.58 | 22.38 | 49.12 | 53.21 | 58.44 | 12.38 | 19.83 | 29.37 | 37.06 |
| **One domain for two clients** | | | | | | | | | | | | |
| CLIP | 11.78 | 6.21 | 9.92 | 8.51 | 9.10 | 11.82 | 9.23 | 9.45 | 10.96 | 10.52 | 9.79 | 10.29 |
| PromptFL | 10.50 | 15.36 | 25.95 | 3.45 | 15.29 | 11.55 | 12.80 | 13.40 | 8.67 | 20.16 | 5.26 | 12.58 |
| FedPGP | 21.02 | 21.38 | 12.70 | 22.07 | 19.29 | 25.61 | 33.52 | 26.51 | 23.62 | 23.62 | 24.89 | 26.30 |
| pFedMoAP | 24.01 | 19.13 | 25.40 | 23.68 | 23.06 | 25.42 | 30.15 | 22.91 | 20.83 | 24.85 | 24.59 | 24.79 |
| pFedMMA (Ours) | 23.73 | 25.88 | 23.67 | 15.00 | 22.07 | 26.61 | 44.42 | 16.94 | 46.09 | 54.11 | 37.46 | 37.61 |
| $\beta = 0.1$ | | | | | | | | | | | | |
| **One domain for one client** | | | | | | | | | | | | |
| CLIP | 10.94 | 8.44 | 0.00 | 13.56 | 8.24 | 11.22 | 9.28 | 10.18 | 11.00 | 9.12 | 10.83 | 10.27 |
| PromptFL | 10.42 | 18.75 | 12.00 | 16.95 | 14.53 | 15.53 | 14.70 | 11.22 | 9.00 | 10.60 | 14.08 | 12.52 |
| FedPGP | 12.50 | 13.33 | 15.63 | 15.25 | 14.18 | 15.59 | 19.18 | 14.38 | 12.00 | 13.15 | 13.00 | 14.55 |
| pFedMoAP | 10.42 | 12.44 | 12.50 | 15.25 | 12.65 | 15.21 | 19.33 | 14.22 | 10.90 | 12.57 | 12.64 | 14.14 |
| pFedMMA (Ours) | 8.93 | 12.33 | 30.16 | 32.89 | 21.08 | 44.19 | 28.35 | 0.00 | 24.03 | 34.33 | 86.21 | 36.18 |
| **One domain for two clients** | | | | | | | | | | | | |
| CLIP | 4.13 | 7.57 | 10.83 | 12.79 | 8.83 | 9.46 | 10.80 | 10.48 | 11.30 | 11.51 | 9.99 | 10.59 |
| PromptFL | 7.13 | 23.75 | 21.34 | 11.94 | 15.99 | 9.63 | 13.07 | 0.50 | 24.98 | 12.89 | 9.89 | 11.83 |
| FedPGP | 18.83 | 21.37 | 25.83 | 24.19 | 22.55 | 24.69 | 32.29 | 28.89 | 19.70 | 24.83 | 26.09 | 26.08 |
| pFedMoAP | 19.41 | 20.96 | 26.67 | 23.90 | 22.73 | 25.60 | 29.66 | 24.85 | 19.70 | 22.84 | 27.32 | 24.99 |
| pFedMMA (Ours) | 13.10 | 30.75 | 29.36 | 14.09 | 21.66 | 50.38 | 38.10 | 60.27 | 61.44 | 40.35 | 46.14 | 49.45 |

## D.3 LEARNING CURVES

To further examine the convergence behavior of pFedMMA, we plot the local accuracy over communication rounds across six representative datasets with five different shot settings in Fig. 3. All methods are evaluated under the same federated setup with 2 local epochs and 50 communication rounds. As shown, pFedMMA consistently achieves high accuracy and exhibits stable, fast convergence across datasets. Notably, even while delivering superior generalization on both base and novel classes (Table 9), pFedMMA converges faster in local performance than pFedMoAP throughout training. These results demonstrate that pFedMMA effectively balances personalization with generalization, ensuring both rapid and reliable convergence.

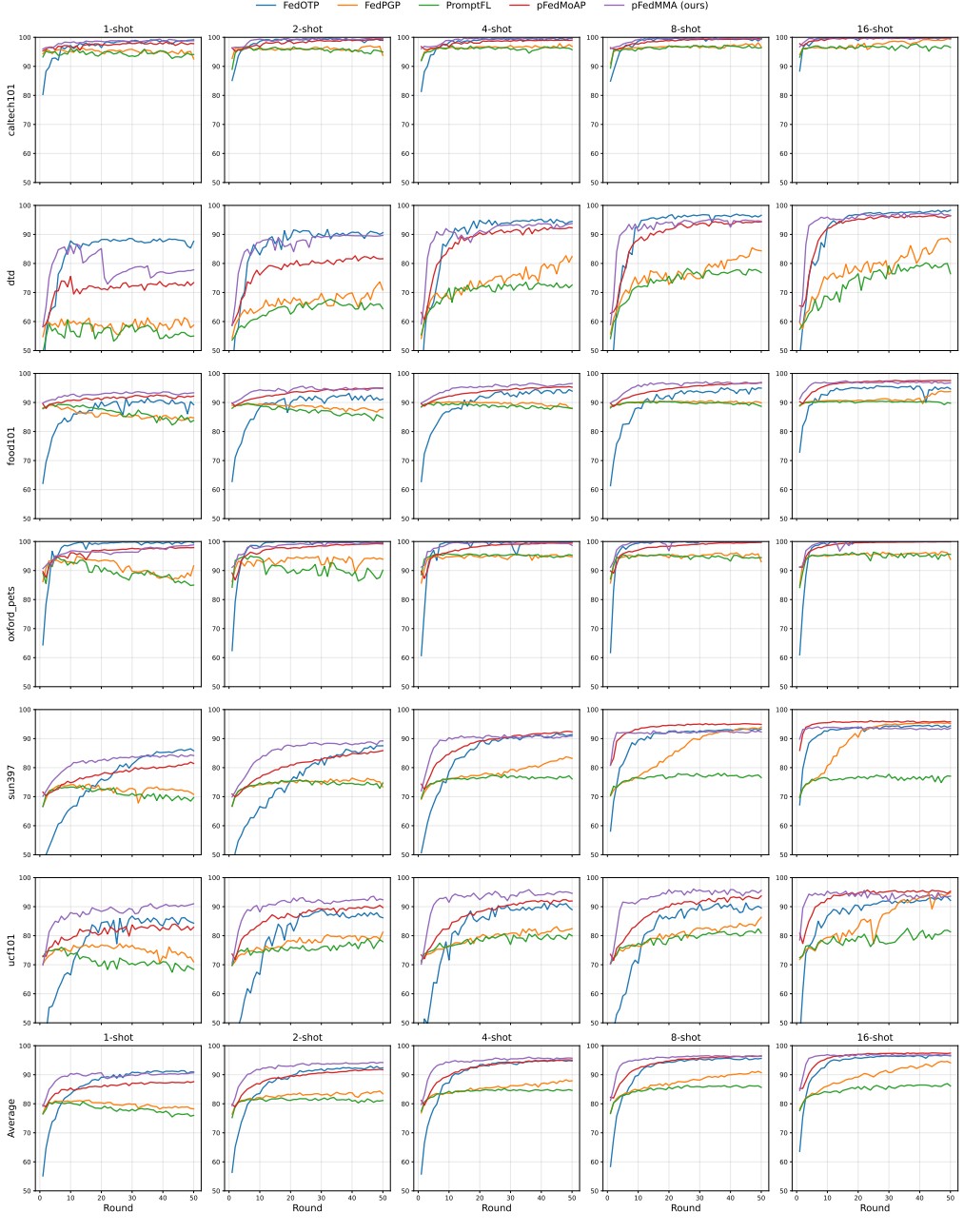

Figure 5: Accuracy learning curves of pFedMMA and baselines over 10 clients.

## D.4 ABLATION STUDY

Table 13: Ablation study on the dimension of the shared adapter for $\alpha = 0.001$.

| Shots | Dimensions | DTD | | | | Caltech101 | | | | UCF | | | | OxfordPets | | | | Average | | | |
|---|---|---|---|---|---|---|---|---|---|---|---|---|---|---|---|---|---|---|---|---|---|
| | | Local | Base | New | HM | Local | Base | New | HM | Local | Base | New | HM | Local | Base | New | HM | Local | Base | New | HM |
| 1 shot | 8 | 66.02 | 58.63 | 62.25 | 62.15 | 98.49 | 94.93 | 92.97 | 95.41 | 83.81 | 63.56 | 68.27 | 70.90 | 93.73 | 62.72 | 84.12 | 77.92 | 85.51 | 69.96 | 76.90 | 76.59 |
| | 16 | 62.50 | 56.62 | 61.97 | 60.24 | 99.30 | 95.19 | 91.66 | 95.28 | 84.11 | 63.20 | 66.90 | 70.32 | 95.58 | 80.97 | 93.84 | 89.63 | 85.37 | 73.99 | 78.59 | 78.87 |
| | 32 | 65.23 | 57.29 | 62.23 | 61.41 | 98.04 | 96.02 | 93.21 | 95.72 | 87.29 | 62.11 | 69.20 | 71.42 | 95.71 | 84.53 | 92.06 | 90.52 | 86.57 | 74.99 | 79.18 | 79.77 |
| | 64 | 65.14 | 54.66 | 63.47 | 60.73 | 98.33 | 94.03 | 92.81 | 95.00 | 85.91 | 60.91 | 67.09 | 69.83 | 93.96 | 81.22 | 91.74 | 88.61 | 85.84 | 72.71 | 78.78 | 78.54 |
| | 128 | 63.29 | 58.61 | 63.16 | 61.61 | 99.21 | 95.00 | 91.98 | 95.31 | 86.67 | 65.35 | 71.02 | 73.31 | 95.11 | 88.19 | 93.83 | 92.28 | 86.07 | 76.79 | 80.00 | 80.63 |
| 2 shots | 8 | 69.72 | 56.09 | 57.51 | 60.53 | 99.63 | 93.89 | 91.06 | 94.73 | 86.11 | 62.40 | 66.67 | 70.36 | 93.41 | 87.15 | 90.74 | 90.36 | 87.22 | 74.88 | 76.50 | 79.00 |
| | 16 | 70.23 | 53.66 | 57.95 | 59.84 | 99.15 | 93.93 | 92.60 | 95.14 | 85.56 | 70.85 | 73.69 | 76.20 | 97.55 | 74.82 | 91.94 | 86.97 | 88.12 | 73.32 | 79.05 | 79.54 |
| | 32 | 70.79 | 47.58 | 61.27 | 58.29 | 99.70 | 94.85 | 91.05 | 95.07 | 91.22 | 67.09 | 70.48 | 74.89 | 94.59 | 85.41 | 92.54 | 90.67 | 89.08 | 73.73 | 78.84 | 79.73 |
| | 64 | 75.05 | 52.25 | 61.04 | 61.42 | 99.46 | 95.25 | 92.77 | 95.75 | 87.75 | 65.18 | 70.62 | 73.35 | 98.66 | 77.67 | 92.07 | 88.57 | 90.23 | 72.59 | 79.13 | 79.77 |
| | 128 | 73.56 | 53.74 | 59.38 | 61.17 | 99.58 | 92.91 | 92.71 | 94.96 | 88.93 | 64.01 | 67.83 | 72.10 | 97.83 | 82.61 | 92.42 | 90.51 | 89.98 | 73.32 | 78.09 | 79.69 |
| 4 shots | 8 | 73.94 | 57.64 | 61.56 | 63.67 | 99.17 | 94.87 | 91.67 | 95.14 | 90.12 | 60.71 | 61.54 | 68.47 | 95.35 | 87.24 | 93.62 | 91.93 | 89.65 | 75.12 | 77.10 | 79.80 |
| | 16 | 73.56 | 55.01 | 49.70 | 57.81 | 99.80 | 93.67 | 90.85 | 94.63 | 90.99 | 60.86 | 65.39 | 70.23 | 96.81 | 86.57 | 94.05 | 92.27 | 90.29 | 74.03 | 75.00 | 78.74 |
| | 32 | 75.83 | 54.36 | 61.40 | 62.67 | 99.64 | 93.89 | 90.31 | 94.46 | 92.69 | 66.70 | 71.86 | 75.57 | 96.99 | 84.15 | 94.52 | 91.54 | 91.29 | 74.78 | 79.52 | 81.06 |
| | 64 | 77.13 | 57.86 | 61.09 | 64.35 | 99.81 | 93.47 | 92.74 | 95.24 | 90.97 | 59.74 | 65.91 | 69.92 | 99.27 | 81.02 | 93.55 | 90.62 | 91.80 | 73.02 | 78.32 | 80.03 |
| | 128 | 81.11 | 52.60 | 57.36 | 61.51 | 99.71 | 88.66 | 90.58 | 92.74 | 92.62 | 60.32 | 68.02 | 71.30 | 98.26 | 81.11 | 93.41 | 90.33 | 92.93 | 70.67 | 77.34 | 78.97 |
| 8 shots | 8 | 76.90 | 56.53 | 54.99 | 61.38 | 99.61 | 95.35 | 91.54 | 95.39 | 89.71 | 57.78 | 63.32 | 67.80 | 95.74 | 81.74 | 93.05 | 89.75 | 90.49 | 72.85 | 75.73 | 78.58 |
| | 16 | 78.10 | 60.06 | 59.89 | 65.00 | 99.74 | 91.15 | 91.34 | 93.91 | 90.26 | 60.18 | 65.65 | 69.88 | 99.59 | 79.80 | 93.80 | 90.27 | 91.92 | 72.80 | 77.67 | 79.77 |
| | 32 | 80.28 | 54.95 | 56.46 | 62.03 | 99.62 | 92.58 | 91.08 | 94.28 | 92.77 | 61.62 | 68.23 | 72.00 | 99.64 | 72.71 | 95.49 | 87.56 | 93.08 | 70.47 | 77.82 | 78.97 |
| | 64 | 85.32 | 55.00 | 57.23 | 63.32 | 99.91 | 93.29 | 92.81 | 95.23 | 91.44 | 54.80 | 66.74 | 67.92 | 99.75 | 77.89 | 92.90 | 89.21 | 94.11 | 70.25 | 77.42 | 78.92 |
| | 128 | 86.62 | 54.97 | 58.77 | 64.17 | 100.00 | 92.11 | 91.34 | 94.33 | 91.44 | 59.83 | 68.00 | 70.83 | 99.53 | 81.98 | 94.76 | 91.47 | 94.40 | 72.22 | 78.22 | 80.20 |
| 16 shots | 8 | 83.52 | 50.22 | 52.61 | 58.95 | 100.00 | 90.99 | 91.90 | 94.13 | 90.50 | 55.48 | 67.47 | 68.34 | 97.58 | 76.65 | 91.27 | 87.59 | 92.90 | 68.34 | 75.81 | 77.25 |
| | 16 | 83.94 | 50.71 | 55.94 | 60.59 | 100.00 | 87.71 | 91.02 | 92.63 | 90.30 | 54.19 | 65.14 | 66.85 | 99.75 | 81.08 | 93.88 | 90.88 | 93.50 | 68.42 | 76.50 | 77.74 |
| | 32 | 85.19 | 47.84 | 56.47 | 59.58 | 99.95 | 86.51 | 89.76 | 91.73 | 93.22 | 57.67 | 65.55 | 69.25 | 99.85 | 76.31 | 93.29 | 88.66 | 94.55 | 67.08 | 76.27 | 77.31 |
| | 64 | 90.79 | 45.31 | 56.97 | 59.24 | 100.00 | 84.25 | 90.88 | 91.26 | 92.46 | 59.04 | 68.03 | 70.67 | 99.85 | 84.44 | 95.66 | 92.85 | 95.78 | 68.26 | 77.89 | 78.51 |
| | 128 | 89.86 | 47.16 | 55.60 | 59.62 | 99.94 | 92.25 | 92.63 | 94.81 | 92.50 | 57.66 | 67.02 | 69.65 | 99.74 | 81.98 | 94.80 | 91.54 | 95.51 | 69.76 | 77.51 | 78.91 |

Table 14: Ablation study on the dimension of the shared adapter for $\alpha = 0.005$.

| Shots | Dimensions | DTD | | | | Caltech101 | | | | UCF | | | | OxfordPets | | | | Average | | | |
|---|---|---|---|---|---|---|---|---|---|---|---|---|---|---|---|---|---|---|---|---|---|
| | | Local | Base | New | HM | Local | Base | New | HM | Local | Base | New | HM | Local | Base | New | HM | Local | Base | New | HM |
| 1 | 16 | 69.03 | 56.49 | 61.21 | 61.82 | 98.48 | 97.19 | 93.84 | 96.46 | 84.67 | 69.74 | 74.52 | 75.82 | 93.47 | 90.30 | 97.19 | 93.57 | 86.41 | 78.43 | 81.69 | 81.92 |
| | 32 | 71.02 | 56.03 | 60.95 | 62.07 | 98.04 | 97.00 | 93.77 | 96.24 | 83.32 | 69.98 | 75.00 | 75.71 | 95.64 | 89.78 | 96.87 | 93.99 | 87.00 | 78.20 | 81.65 | 82.00 |
| | 64 | 66.85 | 55.59 | 59.84 | 60.41 | 97.89 | 96.97 | 94.12 | 96.30 | 84.87 | 70.70 | 74.64 | 76.29 | 93.91 | 90.29 | 96.94 | 93.63 | 85.88 | 78.39 | 81.39 | 81.66 |
| | 128 | 68.98 | 56.31 | 60.39 | 61.46 | 98.90 | 96.95 | 94.10 | 96.61 | 85.01 | 71.20 | 75.78 | 76.91 | 93.76 | 90.00 | 97.22 | 93.57 | 86.66 | 78.61 | 81.87 | 82.14 |
| 2 | 16 | 75.65 | 56.59 | 60.87 | 63.40 | 99.59 | 97.09 | 93.93 | 96.81 | 85.46 | 69.85 | 74.00 | 75.89 | 96.36 | 90.65 | 97.01 | 94.59 | 89.26 | 78.54 | 81.45 | 82.67 |
| | 32 | 71.30 | 56.10 | 61.71 | 62.43 | 99.08 | 97.24 | 93.62 | 96.59 | 88.99 | 70.82 | 75.37 | 77.67 | 96.69 | 90.38 | 96.87 | 94.55 | 89.02 | 78.63 | 81.89 | 82.81 |
| | 64 | 73.52 | 56.24 | 60.70 | 62.69 | 98.92 | 97.08 | 93.76 | 96.54 | 89.64 | 70.71 | 74.58 | 77.51 | 96.22 | 90.14 | 96.62 | 94.23 | 89.57 | 78.54 | 81.42 | 82.74 |
| | 128 | 76.44 | 56.17 | 60.53 | 63.28 | 99.56 | 97.02 | 93.64 | 96.68 | 90.22 | 70.85 | 74.89 | 77.71 | 94.80 | 90.28 | 96.63 | 93.83 | 90.26 | 78.58 | 81.35 | 82.88 |
| 4 | 16 | 81.25 | 57.38 | 61.20 | 65.11 | 99.83 | 97.10 | 94.18 | 96.98 | 90.62 | 70.86 | 74.20 | 77.67 | 96.21 | 90.47 | 96.77 | 94.40 | 91.98 | 78.95 | 81.59 | 83.54 |
| | 32 | 77.69 | 57.80 | 61.81 | 64.72 | 99.74 | 97.20 | 93.88 | 96.88 | 92.76 | 71.51 | 75.02 | 78.75 | 95.51 | 90.07 | 96.63 | 93.98 | 91.42 | 79.14 | 81.83 | 83.58 |
| | 64 | 76.16 | 57.09 | 61.50 | 63.96 | 99.81 | 97.08 | 93.94 | 96.88 | 91.21 | 71.59 | 74.87 | 78.35 | 98.32 | 90.51 | 96.77 | 95.08 | 91.38 | 79.07 | 81.77 | 83.57 |
| | 128 | 81.25 | 57.53 | 61.34 | 65.23 | 99.48 | 97.05 | 93.70 | 96.69 | 92.05 | 71.32 | 74.89 | 78.46 | 96.99 | 90.18 | 96.59 | 94.48 | 92.44 | 79.02 | 81.63 | 83.72 |
| 8 | 16 | 84.26 | 56.56 | 60.95 | 65.28 | 99.57 | 97.05 | 93.93 | 96.79 | 89.85 | 70.20 | 74.49 | 77.32 | 97.11 | 90.79 | 96.73 | 94.42 | 92.70 | 78.40 | 81.53 | 83.45 |
| | 32 | 81.62 | 56.91 | 61.78 | 65.20 | 99.23 | 97.06 | 93.72 | 96.62 | 92.73 | 70.39 | 74.85 | 78.23 | 98.47 | 90.48 | 96.79 | 95.12 | 93.01 | 78.71 | 81.78 | 83.79 |
| | 64 | 81.62 | 56.83 | 60.99 | 64.87 | 99.80 | 97.14 | 93.80 | 96.85 | 91.04 | 71.04 | 74.79 | 78.06 | 98.84 | 90.94 | 96.78 | 95.03 | 92.83 | 78.74 | 81.59 | 83.70 |
| | 128 | 84.63 | 57.01 | 60.91 | 65.54 | 99.83 | 97.04 | 93.79 | 96.82 | 92.66 | 70.38 | 74.92 | 78.23 | 96.99 | 90.20 | 96.67 | 94.51 | 93.53 | 78.66 | 81.57 | 83.78 |
| 16 | 16 | 88.24 | 56.74 | 61.28 | 66.26 | 99.95 | 96.98 | 94.09 | 96.95 | 92.63 | 70.30 | 74.74 | 78.12 | 99.53 | 90.44 | 96.72 | 95.41 | 95.09 | 78.61 | 81.71 | 84.18 |
| | 32 | 86.44 | 56.75 | 61.32 | 65.94 | 99.84 | 97.20 | 93.88 | 96.91 | 92.93 | 69.92 | 74.74 | 78.04 | 99.63 | 90.36 | 96.94 | 95.48 | 94.71 | 78.56 | 81.72 | 84.09 |
| | 64 | 84.03 | 56.25 | 60.56 | 64.95 | 99.95 | 96.87 | 93.60 | 96.74 | 93.28 | 70.16 | 74.45 | 78.11 | 99.10 | 90.06 | 96.67 | 95.12 | 94.09 | 78.34 | 81.32 | 83.73 |
| | 128 | 88.01 | 57.29 | 61.14 | 66.41 | 100.00 | 96.93 | 93.96 | 96.90 | 92.50 | 70.02 | 74.59 | 77.92 | 99.48 | 90.94 | 96.66 | 95.56 | 95.00 | 78.80 | 81.59 | 84.20 |

Table 15: Ablation study on adapter sharing strategies with scaling factor $\alpha = 0.001$, adapter dimension=32, and starting layer $\ell = 5$).

| Shots | Method | Local Performance | | | |
|---|---|---|---|---|---|
| | | DTD | Caltech101 | Flowers102 | OxfordPets |
| 1 shot | No Local Param | 57.41 | 96.04 | 72.27 | 91.75 |
| | Local Shared Adapter | 58.15 | 96.49 | 74.27 | 92.13 |
| | pFedMMA | **64.81** | **98.76** | **80.46** | **95.76** |
| 2 shots | No Local Param | 60.74 | 96.67 | 74.22 | 90.86 |
| | Local Shared Adapter | 60.74 | 98.33 | 78.24 | 91.31 |
| | pFedMMA | **71.25** | **99.41** | **86.74** | **94.60** |
| 4 shots | No Local Param | 60.97 | 96.63 | 73.78 | 91.39 |
| | Local Shared Adapter | 61.72 | 97.85 | 77.30 | 92.82 |
| | pFedMMA | **75.97** | **99.66** | **86.51** | **97.10** |
| 8 shots | No Local Param | 62.13 | 96.97 | 74.90 | 92.32 |
| | Local Shared Adapter | 62.73 | 98.12 | 76.20 | 94.53 |
| | pFedMMA | **80.97** | **99.60** | **86.66** | **99.64** |
| 16 shots | No Local Param | 64.21 | 97.05 | 73.92 | 93.22 |
| | Local Shared Adapter | 71.11 | 99.89 | 79.17 | 92.95 |
| | pFedMMA | **88.89** | **99.95** | **91.32** | **99.85** |

Table 16: Ablation study on scaling factor $\alpha$.

| Shots | Scaling Factor | DTD | | | | Caltech101 | | | | UCF | | | | OxfordPets | | | | Average on 4 datasets | | | |
|---|---|---|---|---|---|---|---|---|---|---|---|---|---|---|---|---|---|---|---|---|---|
| | | Local | Base | Novel | HM | Local | Base | Novel | HM | Local | Base | Novel | HM | Local | Base | Novel | HM | Local | Base | Novel | HM |
| 1 | 0.0001 | 66.06 | 6.74 | 7.10 | 9.86 | 98.52 | 3.51 | 7.33 | 6.95 | 85.08 | 3.12 | 2.53 | 4.12 | 97.19 | 9.73 | 8.36 | 12.89 | 86.71 | 5.78 | 6.33 | 8.46 |
| | 0.0005 | 64.58 | 43.96 | 54.18 | 52.92 | 98.76 | 89.05 | 87.49 | 91.51 | 87.38 | 34.93 | 46.15 | 48.59 | 96.67 | 27.81 | 51.54 | 45.66 | 86.85 | 48.94 | 59.84 | 59.67 |
| | 0.001 | 65.23 | 57.29 | 62.23 | 61.41 | 98.04 | 96.02 | 93.21 | 95.72 | 87.29 | 62.11 | 69.20 | 71.42 | 95.71 | 84.53 | 92.06 | 90.52 | 86.57 | 74.99 | 79.18 | 79.77 |
| | 0.005 | 71.02 | 56.03 | 60.95 | 62.07 | 98.04 | 97.00 | 93.77 | 96.24 | 83.32 | 69.98 | 75.00 | 75.71 | 95.64 | 89.78 | 96.87 | 93.99 | 87.01 | 78.20 | 81.65 | **82.00** |
| | 0.01 | 65.51 | 55.56 | 59.64 | 59.96 | 97.98 | 96.97 | 94.08 | 96.31 | 83.55 | 69.10 | 74.84 | 75.37 | 95.68 | 89.65 | 96.92 | 93.97 | 85.68 | 77.82 | 81.37 | 81.40 |
| 2 | 0.0001 | 76.30 | 4.97 | 5.82 | 7.77 | 99.54 | 5.05 | 2.85 | 5.37 | 89.42 | 2.92 | 2.64 | 4.10 | 95.09 | 8.41 | 14.62 | 15.17 | 90.09 | 5.34 | 6.48 | 8.10 |
| | 0.0005 | 71.30 | 34.05 | 39.61 | 43.71 | 99.64 | 83.46 | 84.57 | 88.65 | 90.40 | 36.51 | 44.09 | 49.07 | 95.08 | 24.92 | 76.76 | 47.12 | 89.11 | 44.74 | 61.26 | 57.14 |
| | 0.001 | 70.79 | 47.58 | 61.27 | 58.29 | 99.70 | 94.85 | 91.05 | 95.07 | 91.22 | 67.09 | 70.48 | 74.89 | 94.59 | 85.41 | 92.54 | 90.67 | 89.08 | 73.73 | 78.84 | 79.73 |
| | 0.005 | 71.30 | 56.10 | 61.71 | 62.43 | 99.08 | 97.24 | 93.62 | 96.59 | 88.99 | 70.82 | 75.37 | 77.67 | 96.69 | 90.38 | 96.87 | 94.55 | 89.02 | 78.63 | **81.89** | **82.81** |
| | 0.01 | 75.56 | 56.30 | 60.82 | 63.24 | 98.99 | 97.13 | 94.00 | 96.66 | 87.04 | 69.60 | 74.93 | 76.53 | 97.34 | 89.93 | 96.88 | 94.59 | 89.73 | 78.24 | 81.66 | 82.76 |
| 4 | 0.0001 | 81.57 | 5.30 | 6.04 | 8.19 | 99.00 | 3.37 | 3.32 | 4.93 | 92.61 | 2.10 | 2.11 | 3.12 | 97.81 | 8.28 | 10.13 | 13.06 | 92.75 | 4.76 | 5.40 | 7.33 |
| | 0.0005 | 78.80 | 38.76 | 44.38 | 49.16 | 99.07 | 73.86 | 73.26 | 80.47 | 93.54 | 28.58 | 35.36 | 40.56 | 95.43 | 54.55 | 78.23 | 72.13 | 91.71 | 48.94 | 57.81 | 60.58 |
| | 0.001 | 75.83 | 54.36 | 61.40 | 62.67 | 99.64 | 93.89 | 90.31 | 94.46 | 92.69 | 66.70 | 71.86 | 75.57 | 96.99 | 84.15 | 94.52 | 91.54 | 91.29 | 74.78 | 80.30 | 81.06 |
| | 0.005 | 77.69 | 57.80 | 61.81 | 64.72 | 99.74 | 97.20 | 93.88 | 96.88 | 92.76 | 71.51 | 75.02 | 78.75 | 95.51 | 90.07 | 96.63 | 93.98 | 91.42 | 79.14 | 81.83 | **83.58** |
| | 0.01 | 80.69 | 56.61 | 60.81 | 64.51 | 99.72 | 97.09 | 93.98 | 96.87 | 89.89 | 69.51 | 74.75 | 77.14 | 97.43 | 90.26 | 96.92 | 94.75 | 91.93 | 78.37 | 81.62 | 83.32 |
| 8 | 0.0001 | 79.03 | 8.19 | 7.58 | 11.25 | 99.51 | 4.15 | 2.21 | 4.26 | 92.47 | 2.69 | 2.94 | 4.15 | 97.63 | 5.52 | 6.43 | 8.65 | 92.16 | 5.14 | 4.79 | 7.08 |
| | 0.0005 | 86.94 | 33.50 | 42.84 | 46.37 | 99.68 | 67.37 | 76.70 | 79.13 | 92.60 | 38.95 | 47.35 | 52.09 | 98.90 | 48.92 | 75.95 | 68.62 | 94.53 | 47.19 | 60.71 | 61.55 |
| | 0.001 | 80.28 | 54.95 | 56.46 | 62.03 | 99.62 | 92.58 | 91.08 | 94.28 | 92.77 | 61.62 | 68.23 | 72.00 | 99.64 | 72.71 | 95.49 | 87.56 | 93.08 | 70.47 | 77.82 | 78.97 |
| | 0.005 | 81.62 | 56.91 | 61.78 | 65.20 | 99.23 | 97.06 | 93.72 | 96.62 | 92.73 | 70.39 | 74.85 | 78.23 | 98.47 | 90.48 | 96.79 | 95.12 | 93.01 | 78.71 | 81.78 | **83.79** |
| | 0.01 | 85.88 | 56.31 | 60.65 | 65.37 | 99.80 | 97.17 | 93.96 | 96.92 | 92.09 | 69.47 | 74.55 | 78.50 | 95.88 | 89.79 | 96.99 | 94.11 | 94.41 | 78.19 | 81.54 | 83.73 |
| 16 | 0.0001 | 90.79 | 6.23 | 8.09 | 10.16 | 99.55 | 2.80 | 4.85 | 5.23 | 91.73 | 2.28 | 2.18 | 3.30 | 99.05 | 5.90 | 7.99 | 9.84 | 95.28 | 4.30 | 5.78 | 7.13 |
| | 0.0005 | 87.36 | 34.47 | 38.74 | 45.27 | 100.00 | 59.99 | 67.04 | 72.14 | 93.04 | 34.73 | 50.00 | 50.38 | 99.85 | 38.44 | 69.02 | 59.38 | 95.06 | 41.91 | 56.20 | 56.79 |
| | 0.001 | 85.19 | 47.84 | 56.47 | 59.58 | 99.95 | 86.51 | 89.76 | 91.73 | 93.22 | 57.67 | 65.55 | 69.25 | 99.85 | 76.31 | 93.29 | 88.66 | 94.55 | 67.08 | 76.27 | 77.31 |
| | 0.005 | 86.44 | 56.75 | 61.32 | 65.94 | 99.84 | 97.20 | 93.88 | 96.91 | 92.93 | 69.92 | 74.74 | 78.04 | 99.63 | 90.36 | 96.94 | 95.48 | 94.71 | 78.56 | 81.72 | **84.09** |
| | 0.01 | 89.49 | 56.28 | 60.57 | 66.00 | 99.89 | 97.08 | 94.01 | 96.93 | 95.63 | 68.93 | 74.79 | 78.26 | 97.30 | 89.68 | 96.99 | 94.52 | 95.58 | 77.99 | 81.59 | 83.93 |

Table 17: Ablation study on scaling factor starting layer $\ell$ with scaling factor $\alpha = 0.005$ and adapter dimension=32.

| Shots | Layers | DTD | | | | Caltech101 | | | | UCF | | | | OxfordPets | | | | Average | | | |
|---|---|---|---|---|---|---|---|---|---|---|---|---|---|---|---|---|---|---|---|---|---|
| | | Local | Base | Novel | HM | Local | Base | Novel | HM | Local | Base | Novel | HM | Local | Base | Novel | HM | Local | Base | Novel | HM |
| 1 | 12 | 90.79 | 54.35 | 60.10 | 65.14 | 99.34 | 97.02 | 94.17 | 96.80 | 91.04 | 69.03 | 75.31 | 77.42 | 99.60 | 89.54 | 96.74 | 95.10 | 95.19 | 77.49 | 81.58 | 83.62 |
| | 10 → 12 | 86.67 | 56.31 | 61.30 | 65.77 | 99.36 | 97.04 | 94.36 | 96.88 | 91.03 | 69.91 | 75.47 | 77.84 | 98.83 | 89.64 | 97.02 | 94.99 | 93.97 | 78.23 | 82.04 | **83.87** |
| | 8 → 12 | 78.38 | 56.20 | 61.06 | 63.93 | 99.34 | 96.89 | 94.17 | 96.75 | 87.60 | 70.09 | 74.45 | 76.70 | 98.40 | 89.91 | 96.94 | 94.94 | 90.93 | 78.27 | 81.66 | 83.08 |
| | 6 → 12 | 68.43 | 55.71 | 60.59 | 61.14 | 99.13 | 97.13 | 94.19 | 96.77 | 81.83 | 70.09 | 75.27 | 75.43 | 93.03 | 89.83 | 96.99 | 93.19 | 85.61 | 78.19 | 81.76 | 81.63 |
| | 5 → 12 | 71.02 | 56.03 | 60.95 | 62.07 | 98.04 | 97.00 | 93.77 | 96.24 | 84.87 | 70.70 | 74.64 | 76.29 | 95.64 | 89.78 | 96.87 | 93.99 | 87.39 | 78.38 | 81.56 | 82.15 |
| 2 | 12 | 89.44 | 53.40 | 59.35 | 64.16 | 99.60 | 97.08 | 93.84 | 96.78 | 94.37 | 68.34 | 75.26 | 77.89 | 98.74 | 89.46 | 96.80 | 95.10 | 95.84 | 76.89 | 81.23 | 83.41 |
| | 10 → 12 | 89.72 | 56.48 | 61.92 | 66.67 | 99.80 | 97.07 | 94.14 | 96.95 | 93.60 | 70.36 | 75.12 | 78.52 | 99.64 | 89.46 | 96.80 | 95.10 | 95.69 | 78.34 | 82.00 | **84.31** |
| | 8 → 12 | 86.90 | 57.12 | 61.76 | 66.36 | 99.62 | 97.19 | 94.00 | 96.90 | 93.34 | 70.66 | 74.88 | 78.49 | 99.80 | 89.80 | 96.57 | 95.10 | 94.93 | 78.69 | 81.80 | 84.24 |
| | 6 → 12 | 76.25 | 56.34 | 61.61 | 63.70 | 99.72 | 97.08 | 93.84 | 96.82 | 90.40 | 69.94 | 74.23 | 77.26 | 95.02 | 89.98 | 96.49 | 93.75 | 90.35 | 78.34 | 81.54 | 82.88 |
| | 5 → 12 | 73.52 | 56.10 | 61.71 | 62.98 | 99.80 | 97.24 | 93.62 | 96.59 | 89.64 | 70.71 | 74.51 | 77.51 | 96.69 | 90.38 | 96.87 | 95.08 | 89.73 | 78.61 | 81.69 | 82.91 |
| 4 | 12 | 92.45 | 53.69 | 59.77 | 64.97 | 99.73 | 96.73 | 94.25 | 96.85 | 94.34 | 68.21 | 74.86 | 77.68 | 100.00 | 88.52 | 96.31 | 94.70 | 96.63 | 76.79 | 81.30 | 83.55 |
| | 10 → 12 | 93.75 | 56.46 | 61.91 | 67.37 | 99.85 | 96.93 | 94.40 | 97.01 | 95.87 | 70.71 | 75.37 | 79.28 | 99.90 | 89.68 | 96.63 | 95.21 | 97.34 | 78.45 | 82.08 | **84.72** |
| | 8 → 12 | 92.82 | 56.67 | 61.67 | 67.21 | 100.00 | 97.08 | 94.17 | 97.02 | 95.93 | 70.69 | 74.98 | 79.14 | 99.79 | 89.81 | 96.92 | 95.32 | 97.14 | 78.56 | 81.94 | 84.67 |
| | 6 → 12 | 78.29 | 57.42 | 61.99 | 64.77 | 99.86 | 97.15 | 93.81 | 96.88 | 92.20 | 71.09 | 74.56 | 78.28 | 99.70 | 89.93 | 97.06 | 95.38 | 92.51 | 78.90 | 81.86 | 83.83 |
| | 5 → 12 | 77.69 | 57.80 | 61.81 | 64.72 | 99.74 | 97.20 | 93.88 | 96.88 | 92.76 | 71.51 | 75.02 | 78.75 | 98.32 | 90.51 | 96.92 | 95.08 | 92.13 | 79.23 | 81.89 | 83.85 |
| 8 | 12 | 94.40 | 53.30 | 59.98 | 65.18 | 99.75 | 96.47 | 94.17 | 96.74 | 94.75 | 68.25 | 75.11 | 77.88 | 100.00 | 88.16 | 95.83 | 94.40 | 97.23 | 76.55 | 81.27 | 83.55 |
| | 10 → 12 | 95.32 | 56.30 | 61.57 | 67.42 | 99.85 | 96.99 | 94.30 | 96.99 | 96.09 | 69.94 | 75.33 | 78.99 | 99.95 | 89.35 | 96.64 | 95.10 | 97.80 | 78.15 | 81.96 | 84.62 |
| | 8 → 12 | 95.00 | 56.90 | 61.46 | 67.61 | 99.98 | 96.97 | 94.07 | 96.95 | 95.76 | 70.51 | 74.96 | 79.02 | 100.00 | 89.46 | 96.99 | 95.27 | 97.69 | 78.46 | 81.87 | **84.71** |
| | 6 → 12 | 85.05 | 57.51 | 61.38 | 66.02 | 99.86 | 97.37 | 93.95 | 97.00 | 90.75 | 70.48 | 75.29 | 77.94 | 99.79 | 90.23 | 97.05 | 95.52 | 93.86 | 78.90 | 81.92 | 84.12 |
| | 5 → 12 | 81.62 | 56.91 | 61.78 | 65.20 | 99.80 | 97.17 | 93.96 | 96.86 | 92.73 | 70.39 | 74.85 | 77.85 | 98.47 | 90.48 | 96.79 | 95.13 | 93.16 | 78.74 | 81.85 | 83.87 |
| 16 | 12 | 94.91 | 53.90 | 60.51 | 65.77 | 99.90 | 95.94 | 94.12 | 96.59 | 95.45 | 68.23 | 75.05 | 78.01 | 100.00 | 88.14 | 96.28 | 94.54 | 97.56 | 76.55 | 81.49 | 83.73 |
| | 10 → 12 | 97.45 | 55.44 | 61.55 | 67.35 | 100.00 | 96.53 | 94.29 | 96.88 | 95.63 | 69.61 | 74.88 | 78.58 | 100.00 | 88.50 | 96.49 | 94.78 | 98.27 | 77.52 | 81.83 | 84.40 |
| | 8 → 12 | 96.30 | 56.66 | 61.21 | 67.61 | 100.00 | 96.72 | 94.19 | 96.91 | 95.86 | 70.14 | 75.11 | 78.94 | 100.00 | 89.18 | 96.77 | 95.10 | 98.04 | 78.18 | 81.82 | **84.64** |
| | 6 → 12 | 88.70 | 56.89 | 61.29 | 66.42 | 99.89 | 97.11 | 94.09 | 96.97 | 92.90 | 69.42 | 74.58 | 77.77 | 99.85 | 89.89 | 96.98 | 95.39 | 95.34 | 78.33 | 81.74 | 84.14 |
| | 5 → 12 | 89.49 | 56.28 | 60.57 | 66.00 | 99.89 | 97.08 | 94.01 | 96.93 | 92.93 | 69.92 | 74.74 | 78.04 | 99.63 | 90.36 | 96.94 | 95.48 | 95.49 | 78.41 | 81.57 | 84.11 |

Table 18: Comparison of FL aggregation variants (vision-only, text-only, and both-sides) for the shared adapter.

| Shots | Layers | Average on 4 datasets | | | | SUN397 | | | | Flowers102 | | | | DTD | | | | Food101 | | | |
|---|---|---|---|---|---|---|---|---|---|---|---|---|---|---|---|---|---|---|---|---|---|
| | | Local | Base | Novel | HM | Local | Base | Novel | HM | Local | Base | Novel | HM | Local | Base | Novel | HM | Local | Base | Novel | HM |
| 16 | Vision Only | 95.81 | 71.19 | 76.07 | 79.31 | 94.02 | 70.73 | 76.08 | 79.12 | 95.79 | 69.62 | 76.31 | 79.14 | 97.31 | 55.22 | 61.10 | 67.04 | 96.13 | 89.17 | 90.80 | 91.94 |
| | Text Only | 95.99 | 71.19 | 76.13 | 79.38 | 94.20 | 70.80 | 76.10 | 79.20 | 96.40 | 69.53 | 76.30 | 79.24 | 97.08 | 55.28 | 61.32 | 67.12 | 96.27 | 89.17 | 90.80 | 91.98 |
| | Both Vision & Text | 95.99 | 71.24 | 76.10 | 79.39 | 94.23 | 70.85 | 76.11 | 79.23 | 96.03 | 69.61 | 76.25 | 79.17 | 97.27 | 55.32 | 61.24 | 67.13 | 96.44 | 89.17 | 90.80 | 92.03 |
| | pFedMMA (Ours) | 96.14 | 71.78 | 76.17 | 79.70 | 94.06 | 70.99 | 76.37 | 79.34 | 95.58 | 71.54 | 76.00 | 79.79 | 97.45 | 55.44 | 61.55 | 67.35 | 97.45 | 89.15 | 90.77 | 92.32 |

Table 19: Top-1 accuracy (%) of different methods across 7 datasets in the 16-shot setting using AdamW optimizer.

| | Average on 7 datasets | | | | SUN397 | | | | Flowers102 | | | | DTD | | | |
|---|---|---|---|---|---|---|---|---|---|---|---|---|---|---|---|---|
| Method | Local | Base | Novel | HM | Local | Base | Novel | HM | Local | Base | Novel | HM | Local | Base | Novel | HM |
| CLIP Radford et al. (2021) | 76.36 | 76.81 | 81.21 | 78.03 | 69.41 | 69.38 | 75.52 | 71.32 | 67.89 | 69.23 | 76.88 | 71.12 | 54.26 | 54.86 | 59.18 | 56.02 |
| PromptFL Guo et al. (2023b) | 86.80 | 86.87 | 79.36 | 84.19 | 77.44 | 77.42 | 73.63 | 76.12 | 89.69 | 89.74 | 73.62 | 83.62 | 76.16 | 75.81 | 54.11 | 66.96 |
| FedPGP Cui et al. (2024) | 97.10 | 63.53 | 67.19 | 73.31 | 95.57 | 41.24 | 49.61 | 54.68 | 99.57 | 47.37 | 57.42 | 61.77 | 95.46 | 48.77 | 44.69 | 56.23 |
| pFedMoAP Luo et al. (2025) | 97.96 | 61.35 | 67.59 | 72.63 | 96.17 | 33.01 | 36.36 | 43.99 | 99.86 | 36.36 | 51.83 | 52.81 | 96.48 | 50.90 | 46.06 | 58.00 |
| pFedMMA (Ours) | 96.29 | 77.90 | 81.65 | 84.58 | 93.78 | 71.17 | 76.51 | 79.40 | 95.79 | 70.93 | 76.85 | 79.89 | 92.82 | 56.91 | 61.50 | 67.26 |

| | OxfordPets | | | | Caltech101 | | | | Food101 | | | | UCF101 | | | |
|---|---|---|---|---|---|---|---|---|---|---|---|---|---|---|---|---|
| Method | Local | Base | Novel | HM | Local | Base | Novel | HM | Local | Base | Novel | HM | Local | Base | Novel | HM |
| CLIP Radford et al. (2021) | 89.45 | 89.42 | 96.81 | 91.77 | 96.14 | 97.22 | 94.21 | 95.84 | 89.40 | 89.42 | 90.70 | 89.84 | 68.00 | 68.15 | 75.18 | 70.29 |
| PromptFL Guo et al. (2023b) | 96.19 | 96.01 | 96.64 | 96.28 | 97.25 | 98.13 | 92.90 | 96.04 | 90.73 | 90.76 | 91.15 | 90.88 | 80.12 | 80.20 | 73.50 | 77.81 |
| FedPGP Cui et al. (2024) | 98.78 | 86.27 | 94.47 | 92.88 | 99.83 | 87.82 | 88.09 | 91.59 | 95.96 | 78.62 | 78.67 | 83.68 | 94.56 | 54.64 | 57.36 | 64.78 |
| pFedMoAP Luo et al. (2025) | 99.90 | 78.13 | 91.76 | 89.00 | 99.94 | 94.62 | 92.47 | 95.58 | 97.60 | 71.43 | 84.37 | 83.11 | 95.76 | 64.98 | 70.26 | 74.88 |
| pFedMMA (Ours) | 99.08 | 89.84 | 96.74 | 95.05 | 100.0 | 97.10 | 94.18 | 97.04 | 97.24 | 89.42 | 90.75 | 92.35 | 95.31 | 69.90 | 75.02 | 78.68 |

# E  TRAINING COST ANALYSIS

## E.1  COMPARISON

Below, we briefly describe each method and provide parametric expressions for the number of trainable parameters and per-round communication, together with instantiations for ViT-B/16; see Table 20 for notation.

**PromptFL.** Each client fine-tunes only a continuous text prompt (the backbone is frozen), and the server aggregates the prompt via FedAvg and broadcasts the updated prompt. (1) *Per-client counts:* The number of trainable parameters is $Ld_t$. In each round, the client uploads $Ld_t$ parameters and downloads $Ld_t$ parameters. (2) *ViT-B/16 example.* With $d_t = 512$ and $L = 16$, one prompt has $Ld_t = 16 \times 512 = 8{,}192$ parameters, so each round the client uploads and downloads 8,192 parameters.

**FedOTP.** Each client learns a global prompt (to be aggregated) and a local prompt (kept private); training couples them via optimal transport, and only the global prompt is communicated. (1) *Per-client counts:* The number of trainable parameters is $2Ld_t$. In each round, the client uploads $Ld_t$ parameters and downloads $Ld_t$ parameters. (2) *ViT-B/16 example.* With $d_t = 512$ and $L = 16$, the client trains $2 \times 16 \times 512 = 16{,}384$ parameters in total, and in each round uploads and downloads 8,192 parameters.

**FedPGP.** Clients share a global prompt and add a low-rank personalized adapter $U_iV_i$ locally; only the global prompt is aggregated. (1) *Per-client counts:* The number of trainable parameters is $Ld_t + b\,(d_t + L)$. In each round, the client uploads $Ld_t$ parameters and downloads $Ld_t$ parameters. (2) *ViT-B/16 example.* With $d_t = 512$, $L = 16$, and $b = 8$, the low-rank component contributes $8(512 + 16) = 4{,}224$ parameters, giving $8{,}192 + 4{,}224 = 12{,}416$ trainable parameters overall; in each round the client uploads and downloads 8,192 parameters.

**pFedMoAP.** Each client learns a local prompt and downloads $K$ non-local prompt experts (without aggregation). A local multi-head attention gating network mixes local and non-local experts; the gating network is trained on-device and not communicated. Features are pooled to width $d_g$ before the MHA. (1) *Per-client counts:* The number of trainable parameters is $Ld_t + (4d_g^2 + 4d_g)$. In each round, the client uploads $Ld_t$ parameters and downloads $K\,Ld_t$ parameters. (2) *ViT-B/16 example.* With $d_t = 512$, $L = 16$, $d_g = 128$, and $K = 9$, the gating network has $4 \cdot 128^2 + 4 \cdot 128 = 66{,}048$ parameters, so the client trains $8{,}192 + 66{,}048 = 74{,}240$ parameters; in each round the client uploads 8,192 parameters and downloads 73,728 parameters.

**pFedMMA.** Lightweight multimodal adapters are inserted in both vision and text blocks; in each instrumented layer the adapter comprises a down-projection ($d \to r$), a shared $r \times r$ projection (aggregated globally), and an up-projection ($r \to d$). The shared projection is communicated each round, while the up/down projections are updated locally. (1) *Per-client counts:* The number of trainable parameters per layer is $2r(d_v + d_t) + r^2$, so across $m$ layers it is $m\,[\,2r(d_v + d_t) + r^2\,]$. In each round, the client uploads $mr^2$ parameters and downloads $mr^2$ parameters. (2) *ViT-B/16 example.* With $d_v = 768$, $d_t = 512$, $r = 32$, and layers 10–12 ($m = 3$), the per-layer trainable count is $2 \cdot 32\,(768 + 512) + 32^2 = 82{,}944$, for a total of $3 \times 82{,}944 = 248{,}832$ trainable parameters; in each round the client uploads and downloads $3 \times 32^2 = 3{,}072$ parameters.

Table 20: Notation used in this part. Examples assume CLIP ViT-B/16.

| Symbol | Description | Example (ViT-B/16) |
|---|---|---|
| $d_t$ | CLIP text-encoder width | 512 |
| $d_v$ | CLIP vision hidden size | 768 |
| $L$ | Number of prompt tokens | e.g., 16 |
| $b$ | Low-rank bottleneck (FedPGP) | e.g., 8 |
| $d_g$ | Internal width of the pFedMoAP gating MHA | e.g., 128 |
| $K$ | Number of non-local prompt experts downloaded per round (pFedMoAP) | e.g., 9 |
| $r$ | Adapter inner (shared) width (pFedMMA) | e.g., 32 |
| $m$ | Number of instrumented transformer layers (pFedMMA) | e.g., 3 |

Table 21 summarizes the computational and communication costs together with accuracy. PromptFL has the smallest footprint (8,192 trainable; 8,192/8,192 per round) but—most importantly—shows a marked drop in local accuracy (88.93%), indicating weaker personalization under heterogeneity. FedPGP increases local trainables to 12,416 without extra communication but incurs the highest memory (13,374 MiB) and slower training, and its HM accuracy (79.09%) lags its strong local accuracy (95.38%). FedOTP doubles prompt capacity (16,384 trainables; same 8,192/8,192 communication) and attains very high local accuracy (97.34%) but suffers extremely low HM accuracy (31.08%), suggesting poor cross-client generalization. pFedMoAP adds a local gating module, raising local trainables (74,240) and the per-round download (73,728) while achieving the shortest training time (902 s) and strong local accuracy (97.89%). pFedMMA (ours) communicates only the shared adapter blocks (3,072/3,072) while keeping 248,832 parameters local, yielding the best HM accuracy (84.15%) and competitive local accuracy (97.17%), thus offering the most favorable accuracy–communication trade-off.

Table 21: Comparison of computation, communication, and accuracy for five personalized federated learning methods under CLIP ViT-B/16. Columns report the number of local trainable parameters, per-round communicated parameters (upload/down), end-to-end training time, peak GPU memory, average local accuracy, and average harmonic-mean (HM) accuracy.

| Methods | # Local Trainable Param. | # Per-round Com. Param. (up/down) | Train Time (s) | GPU Mem. (MiB) | Avg. Local Acc. | Avg. HM Acc. |
|---|---|---|---|---|---|---|
| PromptFL Guo et al. (2023b) | 8,192 | 8,192 / 8,192 | 1,645 | 5,116 | 88.93 | 83.09 |
| FedPGP Cui et al. (2024) | 12,416 | 8,192 / 8,192 | 3,980 | 13,374 | 95.38 | 79.09 |
| FedOTP Li et al. (2024) | 16,384 | 8,192 / 8,192 | 1,328 | 3,014 | 97.34 | 31.08 |
| pFedMoAP Luo et al. (2025) | 74,240 | 8,192 / 73,728 | 902 | 3,108 | 97.89 | 71.05 |
| pFedMMA (Ours) | 248,832 | 3,072 / 3,072 | 2,175 | 4,634 | 97.17 | 84.15 |

## E.2 ON THE NECESSITY OF COMMUNICATING THE SHARED ADAPTER

We ablate adapter sharing to test whether exchanging a small parameter set is sufficient for federated coordination (Table 22). In pFedMMA, only the low-rank *shared* $r \times r$ adapter is globally synchronized each round, while the up/down projections remain local. This design communicates just $mr^2$ parameters per round, yet it is exactly these parameters that carry the essential cross-client signal: they define a common low-dimensional subspace that aligns clients' representations, while the much larger local adapters capture client-specific variation. The comparison with the *Local Only Param* variant (which updates all adapter parameters purely on-device without FL) demonstrates that global synchronization of the shared subspace is crucial, even when the communicated set is small.

Table 22: Ablation study on adapter sharing strategies with scaling factor $\alpha = 0.005$, adapter dimension=32, and starting layer $\ell = 10$).

| Method | DTD | Caltech101 | Flowers102 | OxfordPets | Food | UCF |
|---|---|---|---|---|---|---|
| | | | **Base Performance** | | | |
| Local Only Param | 31.37 | 80.83 | 49.69 | 57.68 | 85.04 | 62.11 |
| pFedMMA | **55.44** | **96.53** | **71.54** | **88.50** | **89.15** | **69.61** |
| | | | **New Performance** | | | |
| Local Only Param | 54.94 | 91.72 | 68.30 | 85.76 | 66.13 | 45.47 |
| pFedMMA | **61.55** | **94.29** | **76.00** | **96.60** | **90.77** | **74.88** |

