# OpenReview forum: "pFedMMA: Personalized Federated Fine-Tuning with Multi-Modal Adapter for Vision-Language Models"
_ICLR.cc/2026/Conference — ICLR 2026 Poster_

### Official Review · Reviewer_742x · 2025-10-22

**Soundness:** 3
**Presentation:** 2
**Contribution:** 2
**Rating:** 4
**Confidence:** 2

**Summary:**

This paper proposes pFedMMA for personalized federated learning (PFL) on vision–language models (VLMs). The core idea is to adopt a Multi-Modal Adapter structure, aggregating only the shared projection on the server while keeping the up/down projections as local personalized parameters. This aims to balance personalization vs. generalization under limited communication cost.

**Strengths:**

* The method is intuitive and easy to implement.
* It has an advantage in terms of communication efficiency.

**Weaknesses:**

1. While the approach is effective for personalized federated VLMs, I have concerns about novelty: the work largely looks like replacing the backbone with a Multi-Modal Adapter in a standard FL pipeline. The substantive difference from prior PEFT/Adapter + FL lines is not sufficiently quantified.
2. The paper emphasizes applicability in out-of-distribution (OOD) scenarios, which is closely related to Federated Domain Generalization (FedDG). However, related work is not discussed in depth and experiments do not compare against FedDG-style algorithms also targeting federated VLMs (e.g., PLAN [1]).

```
Reference
[1] Shuai Gong, Chaoran Cui, Chunyun Zhang, Wenna Wang, Xiushan Nie, and Lei Zhu. Federated domain generalization via prompt learning and aggregation. arXiv:2411.10063, 2024.
```

**Questions:**

1. Why does pFedMMA achieve only 9.26% on the Amazon domain, significantly below FedPGP’s 20.34%? Please provide an explanation/diagnosis.
2. Could you add comparisons of vision-only / text-only / both-sides shared projection to localize the main information-sharing channel and potential side effects?
3. Please include the HM formula and a brief rationale for choosing it in the main text.
4. Fairer baselines beyond prompts: Have you considered other VLM fine-tuning approaches such as CLIP-Adapter + FL (and more general PEFT baselines), given that current baselines are mostly prompt-based?

---

> ### Author Response · Authors · 2025-11-24
>
> We are genuinely grateful for your detailed assessment and valuable insights. Your constructive feedback has significantly contributed to the refinement and advancement of our study.
>
> > W1. While the approach is effective for personalized federated VLMs, I have concerns about novelty: the work largely looks like replacing the backbone with a Multi-Modal Adapter in a standard FL pipeline. The substantive difference from prior PEFT/Adapter + FL lines is not sufficiently quantified.
>
> We appreciate the reviewer’s concern and agree that our FL pipeline (client sampling, local training, weighted aggregation) is deliberately standard, as is common in recent PEFT-for-FL work. The novelty of pFedMMA lies in how we factor and use the multi-modal adapter in the federated setting, rather than in the outer loop itself. Concretely, (i) we introduce a multi-modal adapter tailored to CLIP-style VLMs that explicitly decomposes into client-specific up/down projections and a single shared cross-modal projection, enabling personalization and cross-client knowledge sharing within a unified module; (ii) we propose a federated aggregation scheme that only communicates this shared middle projection, keeping communication comparable to prompt-based methods while offering strictly higher adaptation capacity than prompts; (iii) we ablate which adapter components are updated locally vs.\ globally, showing that our specific choice of local vs.\ shared parts yields the best personalization–generalization trade-off; and (iv) we empirically quantify the effect of this decomposition through new ablations over the information-sharing channel. Importantly, under this design pFedMMA consistently outperforms all state-of-the-art PFL baselines in terms of the harmonic mean of local/base/novel accuracy across several datasets.
>
>
> > W2. The paper emphasizes applicability in out-of-distribution (OOD) scenarios, which is closely related to Federated Domain Generalization (FedDG). However, related work is not discussed in depth and experiments do not compare against FedDG-style algorithms also targeting federated VLMs (e.g., PLAN [1]).
>
> We thank the reviewer for raising this point. Our current paper uses “out-of-distribution” to refer to cross-domain and label-shift generalization among participating clients in a personalized federated learning (PFL) setting, where each domain is a client and maintains its own adapter. This is different from the classical FedDG setup, where a single global model is trained and evaluated on unseen target domains under leave-one-domain-out protocols, as in PLAN. We agree that this distinction was not sufficiently clear and that our discussion of FedDG-style methods was incomplete. In the revised version, we have added a dedicated paragraph on FedDG and PLAN in the related work section, explicitly contrasting their global OOD objective with our per-client PFL objective. We therefore view FedDG approaches such as PLAN as complementary, rather than direct baselines, to our personalized VLM fine-tuning framework.

---

> ### Author Response · Authors · 2025-11-24
>
> > Q1. Why does pFedMMA achieve only $9.26 \%$ on the Amazon domain, significantly below FedPGP's $20.34 \%$ ? Please provide an explanation/diagnosis.
>
>
> We agree that the Amazon result (9.26% vs. 20.34% for FedPGP) is unexpectedly low and merits clarification. In our Office-Caltech10 setup, Amazon is a particularly difficult, small, and atypical domain (web-style product images) and, under the Dirichlet partition with β = 0.5 and only two clients per domain, it suffers from very unbalanced label/domain combinations and strong feature shift. pFedMMA’s multi-modal adapter aggressively adapts both visual and textual branches via the shared projection, which helps on strongly shifted, better-populated benchmarks like DomainNet, but can over-adapt away from CLIP’s prior on such a small, idiosyncratic domain. FedPGP’s prompt-based updates are more conservative and keep the backbone closer to the original model, which appears to be advantageous specifically on Amazon. Importantly, pFedMMA still achieves the best average accuracy across all four Office-Caltech10 domains and substantial gains on Caltech and DSLR; we acknowledge Amazon as an outlier where our stronger adaptation is detrimental.
>
>
> > Q2. Could you add comparisons of vision-only / text-only / both-sides shared projection to localize the main information-sharing channel and potential side effects?
>
> We thank the reviewer for this suggestion. In the revised manuscript, we have added an explicit ablation that varies **which modality-specific shared projections are federated**: **Vision Only** (only the vision-side shared block is aggregated), **Text Only** (only the text-side shared block is aggregated), and **Both Vision & Text** (separate shared blocks for each modality are aggregated simultaneously). As shown in Table 6 below, these variants obtain very similar local accuracy, with small but consistent differences in base, novel, and HM: aggregating the text-side (alone or together with vision) yields a slightly higher HM than aggregating the vision-side alone. Our full pFedMMA, which instead uses a **single multi-modal shared projection** rather than two separate modality-specific ones, further improves local, base, novel, and HM over all three variants, suggesting that tying the modalities through a unified shared adapter provides a stronger and more stable information-sharing mechanism without harming personalization.
>
> | Method              | Local  | Base   | Novel  | HM     |
> |---------------------|--------|--------|--------|--------|
> | Vision Only         | 95.81  | 71.19  | 76.07  | 79.31  |
> | Text Only           | 95.99  | 71.19  | 76.13  | 79.38  |
> | Both Vision & Text  | 95.99  | 71.24  | 76.10  | 79.39  |
> | **pFedMMA (Ours)**  | **96.14** | **71.78** | **76.17** | **79.70** |
>
>
> > Q3. Please include the HM formula and a brief rationale for choosing it in the main text.
>
> We thank the reviewer for this suggestion. In the revised manuscript, we explicitly include the HM formula in the main text:
>
> $$
> HM = 3/(Acc_{local}^{-1} + Acc_{base}^{-1} + Acc_{novel}^{-1}),
> $$
>
> and provide a brief rationale for using HM instead of the arithmetic mean. Specifically, HM penalizes methods that over-optimize one component (e.g., personalization) at the expense of others (base or novel classes), and is therefore better suited to capture the balance we aim for. We also add citations to prior work in generalized zero-shot learning and federated prompt learning that adopt HM to jointly summarize seen/unseen or local/base/novel accuracies, aligning our choice with established practice in the literature.
>
> > Q4. Fairer baselines beyond prompts: Have you considered other VLM fine-tuning approaches such as CLIP-Adapter + FL (and more general PEFT baselines), given that current baselines are mostly prompt-based?
>
> We thank the reviewer for this suggestion. In the revised manuscript, we have added two PEFT-style baselines that go beyond prompt tuning: **FedCLIP-Adapter** and **FedCLIP-LoRA**, where we fine-tune and aggregate only the CLIP-Adapter layers and LoRA layers, respectively, while keeping the VLM backbone frozen and using the same FedAvg protocol as for the prompt-based methods. The results are reported in the updated Table 1. Across all seven datasets in the 16-shot setting, both FedCLIP-Adapter and FedCLIP-LoRA perform competitively with prompt-based PFL methods, but pFedMMA still achieves the best overall harmonic mean (HM) averaged over datasets and consistently stronger base/novel performance on most benchmarks. These new experiments address the concern about fairness by showing that our gains are not specific to a prompt-only comparison, but also hold against adapter- and LoRA-based federated fine-tuning baselines. We will add the corresponding results for other datasets and shot settings in the final manuscript.

---

### Official Review · Reviewer_quYy · 2025-10-26

**Soundness:** 3
**Presentation:** 3
**Contribution:** 2
**Rating:** 4
**Confidence:** 4

**Summary:**

The paper proposes pFedMMA, a novel personalized federated learning (PFL) framework for fine-tuning vision-language models (VLMs) like CLIP in decentralized, heterogeneous settings. It introduces multi-modal adapters with modality-specific down- and up-projection layers and a shared projection layer, inserted into upper transformer blocks of both image and text encoders. Clients locally update all adapter components to adapt to their data distributions, while only the shared projection is globally aggregated via FedAvg, balancing personalization and generalization with communication efficiency.

**Strengths:**

Comprehensive Evaluation: The study spans diverse heterogeneity scenarios (label shifts via non-overlapping classes, feature shifts via multi-domain datasets like DomainNet), using Dirichlet partitioning for realistic non-IID data. Testing extensive datasets, two backbones (ViT-B/16, ViT-B/32), and few-shot regimes, provides robust evidence of applicability and interpretability.

Efficiency and Scalability: As a parameter-efficient fine-tuning (PEFT) method, it freezes the VLM backbone, training only lightweight adapters. The focus on shared projection aggregation reduces communication costs.

**Weaknesses:**

Unverified Cross-Modal Alignment: The core claim of achieving cross-modal consistency via the shared projection lacks rigorous validation. The parallel adapter design with a shared layer assumes modality interaction without explicit mechanisms (e.g., attention or fusion gates), and no quantitative evidence (e.g., cosine similarity, t-SNE visualizations) confirms reduced modality gaps or alignment under federated heterogeneity.

Insufficient Motivation and Problem Framing: The motivation relies on prompt-based PFL methods, sacrificing generalization for personalization, but lacks deep analysis on why adapters inherently outperform prompts or why a hybrid prompt-adapter approach is not explored. Baseline selection is biased toward prompt methods, omitting adapter-based methods such as FedCLIP (Lu et al., 2023).

**Questions:**

Please see the weaknesses above.

---

> ### Author Response · Authors · 2025-11-24
>
> We deeply appreciate your thorough review of our manuscript and are grateful for the insightful feedback you provided. Your constructive comments have been instrumental in enhancing the quality of our work.
>
> > Unverified Cross-Modal Alignment: The core claim of achieving cross-modal consistency via the shared projection lacks rigorous validation. The parallel adapter design with a shared layer assumes modality interaction without explicit mechanisms (e.g., attention or fusion gates), and no quantitative evidence (e.g., cosine similarity, t-SNE visualizations) confirms reduced modality gaps or alignment under federated heterogeneity.
>
> We thank the reviewer for this comment. Our intention was not to claim that we have rigorously *measured* a reduction of the modality gap, but rather to describe an architectural bias: the shared projection is jointly updated from image–text contrastive losses and is reused in both branches, which encourages the two modalities to adapt in a coordinated way. We agree that, in the current version, phrases such as “cross-modal consistency” may suggest a stronger, feature-level alignment claim than what we actually validate.
>
> In this work, our primary objective is to improve **task-level performance** in the federated setting, i.e., local/base/novel accuracy and their harmonic mean—rather than to explicitly minimize a modality-gap metric. We therefore focus our empirical analysis on accuracy-based evaluations and FL-specific ablations (e.g., which adapter parts are local vs.\ shared, and which modality-specific shared blocks are aggregated), which directly reflect the effectiveness of the proposed design for personalized and cross-domain generalization. While feature-space diagnostics such as cosine similarity or t-SNE are certainly interesting, we believe they are somewhat orthogonal to the main PFL story and would not change the core claim of the paper, which is about improved performance under heterogeneity rather than provably reduced modality gaps.

---

> ### Author Response · Authors · 2025-11-24
>
> > Insufficient Motivation and Problem Framing: The motivation relies on prompt-based PFL methods, sacrificing generalization for personalization, but lacks deep analysis on why adapters inherently outperform prompts.
>
> We appreciate this concern and have clarified our motivation accordingly. Our goal is not to claim that adapters inherently and universally outperform prompts, but to argue that, for federated personalization of VLMs, adapter-style PEFT offers a more expressive and stable way to adapt CLIP than prompt-only tuning.
>
> CLIP-Adapter [1] shows that inserting small bottleneck adapters into CLIP’s feature space can consistently outperform strong prompt-tuning baselines (e.g., CoOp) across multiple few-shot classification benchmarks while keeping the backbone frozen, and a series of continual / multi-task CLIP works further demonstrate that adapter-based tuning largely avoids catastrophic forgetting by freezing the backbone and updating only lightweight residual modules (often via task-specific or MoE-style adapters) instead of the full network [2,3]. Together, these results suggest that adapter-style PEFT can be both more effective and more stable than pure prompt tuning for CLIP-like VLMs, especially under distribution shift.
>
> In our PFL formulation, each client has very limited, biased data; purely local prompt tuning can overfit the input space and distort CLIP’s zero-shot geometry, while per-client adapters in intermediate layers let us (i) keep the raw text/image tokenization untouched and (ii) share a small cross-modal adapter globally to regularize local updates. We now emphasize that this is a hypothesis supported by both prior PEFT results and our ablations (where pFedMMA improves HM over prompt-based PFL baselines).
>
> [1] CLIP-Adapter: Better Vision-Language Models with Feature Adapters(Gao et al., 2021)
>
> [2] Boosting Continual Learning of VLMs via Mixture-of-Experts Adapters (Yu et al., CVPR 2024)
>
> [3] LADA: Scalable Label-Specific CLIP Adapter for Continual Learning (Luo et al., ICML 2025)
>
> > Why a hybrid prompt-adapter approach is not explored.
>
> We appreciate the reviewer’s suggestion. Our goal in this work is to isolate and study the effect of a multi-modal adapter in the PFL setting, with a clear factorization between client-specific and shared components. Introducing hybrid prompt–adapter designs would add several extra design choices (e.g., where to place prompts vs. adapters, how to jointly aggregate them, and how to balance their capacities), making it harder to attribute gains specifically to the proposed multi-modal adapter and its aggregation scheme. To keep the scope focused and the analysis interpretable, we therefore restrict pFedMMA to an adapter-only architecture, while comparing against strong prompt-based PFL baselines as well as adapter/LoRA baselines (FedCLIP-Adapter, FedCLIP-LoRA). We agree that hybrid prompt–adapter PFL is an interesting direction and explicitly view it as orthogonal, follow-up work building on the present study.
>
>
> > Baseline selection is biased toward prompt methods, omitting adapter-based methods such as FedCLIP (Lu et al., 2023).
>
> We thank the reviewer for this comment. In the revised manuscript, we have strengthened the baseline set beyond prompt-based methods by adding two PEFT-style adapter baselines: **FedCLIP-Adapter** and **FedCLIP-LoRA**, where only adapter or LoRA layers are fine-tuned and aggregated in FL. As reported in the updated Table 1, pFedMMA still achieves the best harmonic mean over local/base/novel accuracy, indicating that our gains are not an artifact of comparing only to prompt-based PFL.
>
> We also agree that FedCLIP-style works are relevant, but they typically target a somewhat different goal and protocol—namely **federated domain/OOD generalization with a single global model**, rather than personalized FL where each client maintains its own model or adapter. In the revised related work section, we have added a dedicated paragraph on federated OOD/domain generalization methods (including FedCLIP-style approaches and FedDAT), and we explicitly position them as a complementary line of work with a different objective from our PFL setting.

---

### Official Review · Reviewer_Tmrt · 2025-10-28

**Soundness:** 3
**Presentation:** 2
**Contribution:** 2
**Rating:** 4
**Confidence:** 4

**Summary:**

This paper proposes pFedMMA, a personalized federated learning framework for Vision-Language Models (VLMs), which for the first time incorporates multi-modal adapters into federated fine-tuning. The proposed architecture includes modality-specific up- and down-projection layers along with a globally shared projection layer. During training, all components are updated locally, while only the shared projection is aggregated globally. Extensive experiments across multiple datasets demonstrate that pFedMMA achieves a SOTA trade-off between personalization and generalization.

**Strengths:**

1. pFedMMA effectively introduces multi-modal adapters into personalized federated learning, balancing personalization and generalization. It addresses the poor generalization of existing prompt-tuning methods on unseen classes.

2. The asymmetric training mechanism, which aggregates only the shared projection layer, reduces communication costs while retaining modality-specific up- and down-projections locally to adapt to local data distributions.

3. Through extensive evaluation across diverse data heterogeneity scenarios, pFedMMA is shown to surpass prior prompt-based PFL techniques in generalization across both domains and categories, without compromising its personalization strength.

**Weaknesses:**

1. Although communication cost is reduced, the total number of trainable parameters introduced by pFedMMA is significantly larger than mainstream prompt-tuning methods, increasing local computational and memory burdens, which may not be friendly to resource-constrained devices.

2. Despite achieving the best harmonic mean (HM) performance, pFedMMA shows noticeably lower local accuracy than pFedMoAP on several datasets (e.g., Flowers102 and DTD), indicating that its personalization capability is sacrificed in certain scenarios. The overall performance is sensitive to dataset distributions and lacks stability.

3. In the domain generalization experiments on DomainNet and Office-Caltech10, the experiments do not include federated baselines explicitly developed for domain or feature shift scenarios, which weakens the credibility of their claims regarding domain generalization capability.

**Questions:**

1. The paper claims that the shared projection layer improves generalization to unseen classes, but it does not explain why this structure effectively generalizes to semantic categories completely absent during training.

2. The motivation for using harmonic mean (HM) as the main evaluation metric, rather than arithmetic mean, is not sufficiently justified. Moreover, no references are provided to support the use of HM for evaluating the balance between personalization and generalization.

3. The main contribution appears to be a direct adaptation of the centralized MMA to the federated setting, with the added strategy of aggregating only the shared projection layer globally. This raises concerns about limited novelty.

---

> ### Author Response · Authors · 2025-11-24
>
> We sincerely thank you for your careful reading of our paper and appreciate the valuable feedback in your comments. The insightful and constructive suggestions have enabled us to effectively improve our work.
>
> > W1. Although communication cost is reduced, the total number of trainable parameters introduced by pFedMMA is significantly larger than mainstream prompt-tuning methods, increasing local computational and memory burdens, which may not be friendly to resource-constrained devices.
>
> We appreciate the reviewer’s observation. Compared to prompt-tuning methods, pFedMMA indeed introduces more trainable parameters per client, since we add modality-specific up/down projections instead of a few prompt vectors. This is an intentional trade-off: we increase local adaptation capacity while still avoiding the communication and memory cost of full model fine-tuning. The adapter parameters remain a small fraction of the frozen VLM backbone, and only the shared middle projection is communicated, keeping communication comparable to prompt-based PFL. The extra computation comes from narrow bottleneck layers and is modest relative to the overall transformer cost. We agree that for extremely resource-constrained devices, prompt-based PFL may remain more suitable.
>
>
> > W2. Despite achieving the best harmonic mean (HM) performance, pFedMMA shows noticeably lower local accuracy than pFedMoAP on several datasets (e.g., Flowers102 and DTD), indicating that its personalization capability is sacrificed in certain scenarios. The overall performance is sensitive to dataset distributions and lacks stability.
>
> We agree that pFedMMA can have slightly lower local accuracy than pFedMoAP on a few benchmarks (e.g., Flowers102, DTD). This is a consequence of our design goal: pFedMMA is explicitly more “global-regularized,” trading a bit of pure personalization for much stronger base/novel and HM performance across datasets. In other words, we prioritize the overall personalization–generalization balance rather than maximizing local accuracy alone. In our experiments, this behavior is consistent across runs with fixed hyperparameters, so we do not observe instability, but we acknowledge that the optimal trade-off is dataset-dependent.
>
> > W3. In the domain generalization experiments on DomainNet and Office-Caltech10, the experiments do not include federated baselines explicitly developed for domain or feature shift scenarios, which weakens the credibility of their claims regarding domain generalization capability.
>
> We appreciate the reviewer’s concern. Our experiments on DomainNet and Office-Caltech10 are conducted in the personalized federated learning setting, where all domains participate as clients and each client maintains its own adapter; our “domain generalization” metrics refer to cross-domain / cross-client performance within this PFL protocol, rather than the classical FedDG setting with held-out unseen target domains. Federated domain generalization methods (including FedDAT, PLAN, etc.) are designed to optimize a single global model under leave-one-domain-out or unseen-target-domain protocols, which makes a direct comparison not entirely aligned with our per-client PFL objective. We have clarified this distinction in the revised paper by adding a dedicated paragraph on FedDG/OOD methods in the related work and by explicitly describing our evaluation as cross-domain generalization within a PFL setting, where we show that our approach achieves comparable personalization while substantially improving generalization performance over existing personalized VLM baselines.

---

> ### Author Response · Authors · 2025-11-24
>
> > Q1. The paper claims that the shared projection layer improves generalization to unseen classes, but it does not explain why this structure effectively generalizes to semantic categories completely absent during training.
>
> We thank the reviewer for raising this point. Our claim is not that the shared projection creates new semantic knowledge, but that it helps pFedMMA better preserve and reuse the semantic prior already encoded in the pre-trained CLIP model. In our base/novel protocol, novel classes never appear during federated training, but their text prompts are still mapped by CLIP’s frozen text encoder, which already provides a class-agnostic image–text alignment learned from large-scale pre-training. Our multi-modal adapter is a small residual module on top of this backbone, and the shared projection, trained on local and base classes across all clients, learns generic cross-modal corrections in the same feature space, which naturally transfer to unseen classes. In contrast, purely local adapters or prompts can overfit to each client’s limited class/domain distribution and partially distort CLIP’s base-to-novel behavior. The shared projection acts as a regularizer that keeps the adapted representation closer to CLIP’s globally consistent geometry while still allowing client-specific up/down projections to personalize.
>
> > Q2. The motivation for using harmonic mean (HM) as the main evaluation metric, rather than arithmetic mean, is not sufficiently justified. Moreover, no references are provided to support the use of HM for evaluating the balance between personalization and generalization.
>
> We appreciate the reviewer’s comment. In the revised manuscript, we now (i) explicitly introduce the HM formula in the main text
> $$
> HM = 3/(Acc_{local}^{-1} + Acc_{base}^{-1} + Acc_{novel}^{-1}),
> $$
>  and (ii) clarify why we prefer HM over the arithmetic mean in our setting. Since our goal is to balance **personalization** (local accuracy) and **generalization** (base and novel accuracies), HM is more appropriate because it penalizes imbalanced solutions that over-optimize one component at the expense of the others, and thus better reflects the overall trade-off we care about. We also added references to prior work where HM is standard for jointly evaluating seen vs.\ unseen or base vs.\ novel performance in generalized zero-shot learning and CLIP adaptation, as well as recent federated prompt learning methods that report HM over local/base/novel accuracies, aligning our choice with established practice in the literature.
>
>
> > Q3. The main contribution appears to be a direct adaptation of the centralized MMA to the federated setting, with the added strategy of aggregating only the shared projection layer globally. This raises concerns about limited novelty.
>
> We appreciate the reviewer’s concern. While our design is inspired by centralized MMA, pFedMMA is not just “MMA + FedAvg.” MMA is trained as a single centralized model, whereas we explicitly re-factor the adapter for PFL: (i) we split it into client-specific up/down projections and a shared cross-modal projection, (ii) we design a partial aggregation scheme that only communicates this shared block (keeping communication comparable to prompts), and (iii) we provide FL-specific ablations on which adapter parts are local vs.\ global and which information-sharing channel (local-only / vision-only / text-only / both) is used. Importantly, under this design pFedMMA consistently outperforms all state-of-the-art PFL baselines in terms of the harmonic mean of local/base/novel accuracy across several datasets, showing that these architectural choices translate into concrete gains beyond a naive MMA-to-FL adaptation.

---

> > ### Comment · Reviewer_Tmrt · 2025-11-28
> >
> > Thank you for the authors' response. I remain skeptical about the novelty of the paper, but most of my concerns have been addressed. Therefore, I will raise my score to 6.

---

### Official Review · Reviewer_VEYb · 2025-11-05

**Soundness:** 4
**Presentation:** 3
**Contribution:** 2
**Rating:** 6
**Confidence:** 4

**Summary:**

The paper designs an algorithm (pFedMMA) to personalize Vision-Language Models in a federated learning setup to adapt to client-side data heterogeneity. The design is based on client-specific multi-modal parallel adapters with each adapter's cross-modal shared projection layer aggregated for generalization under federated learning. The paper presents extensive empirical evidence (under data heterogeneity, label-shift, and feature-shift) that pFedMMA improves the generalization-personalization trade-off compared to personalized federated prompt tuning methods. The adapters are restricted to higher layers of the image and text encoders of the VLM to keep communication cost of aggregating the shared projection layers in check while capturing a large proportion of the full potential of the design w.r.t. generalization and personalization.

**Strengths:**

(S1) The paper is very well-organized and well-written. It is delightfully easy to read. Intuitions from related work are provided in several places for proper contextualization. The problem, algorithm design, and experimental setups are all well motivated. Results and intuition are well communicated.

(S2) Experiments and metrics presented provide good quality empirical evidence to support the claims in the paper. A diversity of datasets and experimental conditions covering data heterogeneity, label-shift, and feature-shift have been considered. The results communicate that a demonstrated improvement is achieved by pFedMMA in the generalization-personalization trade-off in federated learning of VLMs for several datasets.

**Weaknesses:**

(W1) While related work is mostly well cited, I believe that one relevant paper [FedDAT] is missing. Contributions of this manuscript should be contextualized and differentiated w.r.t. this reference. [FedDAT] Chen, H., Zhang, Y., Krompass, D., Gu, J., & Tresp, V. (2024). FedDAT: An Approach for Foundation Model Finetuning in Multi-Modal Heterogeneous Federated Learning. Proceedings of the AAAI Conference on Artificial Intelligence, 38(10), 11285-11293.

(W2) Lines 461-462: Based on Fig. 4, one can only comment about evolution of personalization accuracy, not generalization. To support the analogous generalization claim, a plot of the base-to-novel generalization vs communication round is needed.

(W3) Line 357: It would be good to have results with one more optimizer (Muon or Adam) besides SGD to ensure that the conclusions hold true on change of optimizer. Results can be in the appendix, but referenced & commented on in the main body.

Things to improve the paper that did not impact the score:
- Lines 363-365: It would be good to have a cited reference for the use of "base-to-novel generalization" metric.
- Section 4.3: Among all the studies presented, only the "Adapting Variant Options for PFL" is an ablation study. The others are hyperparameter choice experiments.

**Questions:**

(Q1) Lines 18-20 - "In this work, we propose pFedMMA, the first personalized federated learning framework that leverages multi-modal adapters for vision-language tasks." I feel that this is a bit too strong of a claim, since several elements of the design already appear in the related works - a) [FedDAT] incorporates adapters in FL, and b) FedPGP deals with generalization-personalization trade-offs. Is it more apt to state this work as an advancement (with explicit clarifications on what parts are advancements)? Same comment for Lines 86-88.

(Q2) Would pFedMMA translate well to state-of-the-art privacy and security aware enhancements to the aggregation method? Some explicit commentary on this aspect is warranted in the paper's main body (or in the appendix and referenced in the main body).

(Q3) Lines 264-265: Should this aggregation formula be changed when number of samples varies across clients? How are the empirical results generated for large differences in number of samples?

---

> ### Author Response · Authors · 2025-11-24
>
> We sincerely thank you for your comprehensive examination of our paper and value the thoughtful feedback you have offered. Your helpful suggestions have played a crucial role in improving the overall quality of our research.
>
> > **W1: Missing FedDAT in Related Work**
>
> FedDAT belongs to the line of federated OOD / domain generalization methods, where the objective is to fine-tune a single global multi-modal foundation model that is robust to domain shift across clients. In contrast, our work is formulated in the personalized federated learning setting: each client maintains its own adapter and we explicitly optimize the personalization–generalization trade-off, reporting per-client performance as well as cross-domain accuracy among participating clients. We have added a paragraph in the related work section discussing FedDAT and other FedDG approaches, and clarifying that we view them as complementary rather than direct baselines to our personalized VLM fine-tuning framework.
>
>
> >  **W2: Insufficient Evidence for Generalization In Fig 4**
>
> We thank the reviewer for pointing this out. We agree that Fig. 4, as originally presented, only supports conclusions about the evolution of **personalization accuracy** across communication rounds. In the revised manuscript, we have corrected the text around Lines 461–462 explicitly state that the curve tracks *personalized (local) accuracy* over rounds. We no longer claim that Fig. 4 shows the dynamics of base-to-novel generalization. Instead, base and novel accuracies are reported as standard test-time metrics in our main tables (i.e., evaluated on fixed test sets after training), and we focus the temporal plot on personalization accuracy, which is the quantity that is meaningfully defined per round in our current evaluation protocol.
>
> > **W3: Robustness to Optimizer Choice**
>
> We thank the reviewer for this suggestion. In addition to the main experiments with SGD, we have run pFedMMA and the baselines using the Adam optimizer and reported the full results in the Appendix (Table 18). The conclusions remain consistent: pFedMMA continues to achieve the best overall HM under Adam, indicating that our findings are robust to the choice of optimizer. In the final version, we will also add Adam results for additional datasets and shot configurations to further support this robustness claim.
>
> > **Q1: Overstated “First” Claim for pFedMMA**
>
> We thank the reviewer for raising this concern. In the revised manuscript, we have removed the strong “first” wording in both the abstract and the introduction and now describe pFedMMA as an adapter-based framework for personalized federated learning of vision–language models, with clear clarification of what is new. At the same time, we would like to emphasize our specific novelty: to the best of our knowledge, this is the first work that brings a *multi-modal adapter* architecture with a shared cross-modal projection into the **personalized** FL setting for VLMs. Existing multi-modal adapter methods such as MMA are trained centrally (not federated), while FL methods with adapters such as FedDAT focus on a single global model rather than per-client personalization; conversely, personalized federated VLM methods like FedPGP operate at the prompt level and do not use multi-modal adapters. Our contribution is to combine these lines by (i) introducing a multi-modal adapter with modality-specific up/down projections and a shared cross-modal block and (ii) designing a PFL training/aggregation scheme that communicates only this shared block while keeping the rest of the adapter personalized.
>
> > **Q2: Compatibility with Privacy-Aware Aggregation**
>
> We thank the reviewer for this suggestion. pFedMMA builds on a standard FedAvg-style aggregation loop and only changes *which* parameters are communicated (a small shared cross-modal adapter block), not the aggregation mechanism itself. This design is naturally compatible with existing privacy- and security-aware enhancements (e.g., differential privacy or secure aggregation), and the reduced number of communicated parameters can, in principle, translate into stronger privacy–utility trade-offs under DP-style mechanisms. However, a thorough investigation of these aspects is orthogonal to our current focus on personalization and cross-domain generalization and is therefore beyond the scope of this work; we regard it as an interesting and important direction for future research.
>
> > **Q3: Sample-Size Weighting in Aggregation Formula**
>
> All experiments use the following aggregation formula:b$W_{j s}^{t+1} = \sum_{i=1}^{N} p_i W_{j s, i}^{t,E},$ where $p_i = \frac{n_i}{n}$ scales the contribution of each client by its dataset size $n_i$, with $n = \sum_i n_i$ and $N$ the number of participating clients.
>
> In lines 264–265, we had (incorrectly) assumed that all clients have the same number of samples. This has been corrected in the revised manuscript (lines 264–265) and in Algorithm 1.

---

### Author Response · Authors · 2025-12-03
**Final Remarks**

Dear Area Chair,

We thank all reviewers for their time and constructive feedback. We appreciate that they highlighted the overall clarity and organization of the paper and the quality of the problem setup and intuitions (Reviewer VEYb, quYy), the methodological contribution of bringing multi-modal adapters into personalized FL for VLMs and explicitly targeting the personalization–generalization trade-off (Reviewer Tmrt, quYy, 742x), the breadth and robustness of the empirical evaluation across diverse heterogeneity scenarios and backbones (Reviewer VEYb, Tmrt, quYy), and the efficiency and practicality of our design, including parameter-efficient fine-tuning and reduced communication via shared projection aggregation (Reviewer Tmrt, quYy, 742x).

Below, we briefly summarize the key revisions.

- **Baselines (prompts and adapters).**
We added federated CLIP-Adapter and CLIP-LoRA as adapter-based FL baselines to the main 7-dataset, 16-shot table (Table 1); pFedMMA still achieves the best HM while remaining competitive in local accuracy, showing gains over both prompt- and adapter-based methods.

- **Aggregation variants ablation.**
We introduced a new ablation comparing vision-only, text-only, and both-sides aggregation against our unified shared projection (Lines 471-485). All shared variants improve base/novel/HM over purely local training, and our unified multi-modal shared adapter gives the strongest HM without hurting personalization.

- **Optimizer robustness.**
We added an Adam version of the 7-dataset, 16-shot experiment in the appendix (Table 18) and reference it in the main body (Lines 354-355); the results confirm that pFedMMA continues to obtain the best HM under Adam as well as SGD.

- **OOD / FedDG setting and related work.**
We added a concise related-work paragraph on federated OOD / FedDG, covering FedDAT-style multi-modal FL and FedDG methods such as PLAN and other adapter/prompt-based approaches that train a single global model for unseen domains (Lines 955-971). We clarify that these works target global domain generalization, whereas pFedMMA is formulated in a personalized FL setting with per-client models and Local/Base/Novel evaluation, so we position them as complementary rather than primary baselines.

- **Additional clarifications.**
We clarified that aggregation is weighted by client sample counts (Lines 263-264).We now include the explicit harmonic mean formula over Local/Base/Novel accuracies in the main text, explain that HM captures balance rather than just average performance, and keep reporting all four metrics (Local, Base, Novel, HM) for every method (Lines 364-369).

Warm regards,

Authors of submission 11070

---

### Meta-Review · Area_Chair_MG1b · 2026-01-06

**Summary:**

Reviewers’ key concerns: Missing related work (FedDAT, FedDG/PLAN, FedCLIP); optimizer robustness; high compute burden; cross-modal alignment; novelty doubts; unclear HM rationale; Amazon domain underperformance; biased baselines. Authors addressed most (added adapter/LoRA baselines, HM formula, ablations, related work clarifications)

**Reviewer Concerns:**

no outstanding concerns other than novelty (Reviewer Tmrt)

**Reviewer Scores:**

Reviewer Tmrt may raise his score as discussed in the rebuttal.

---

### Decision · Program_Chairs · 2026-01-26

Accept (Poster)